# Optimal Bayesian Stopping for Efficient Inference of Consistent LLM Answers

**Jingkai Huang** [1]   **Will Ma** [2]   **Zhengyuan Zhou** [1]

## Abstract

A simple strategy for improving LLM accuracy, especially in math and reasoning problems, is to sample multiple responses and submit the answer most consistently reached. In this paper we leverage Bayesian prior information to save on sampling costs, stopping once sufficient consistency is reached. Although the exact posterior is computationally intractable, we further introduce an efficient "$L$-aggregated" stopping policy that tracks only the $L-1$ most frequent answer counts. Theoretically, we prove that $L = 3$ is all you need: this coarse approximation is sufficient to achieve asymptotic optimality, and strictly dominates prior-free baselines, while having a fast posterior computation. Empirically, this identifies the most consistent (i.e., mode) LLM answer and achieves similar answer accuracy using fewer samples.

## 1. Introduction

Large Language Models (LLMs) have demonstrated remarkable capabilities in complex reasoning tasks. A key catalyst for this success is Chain-of-Thought (CoT) prompting, which encourages models to generate intermediate reasoning steps before deriving a final answer (Brown et al., 2020; Kojima et al., 2022; Wei et al., 2022). However, a single decoding pass remains prone to hallucinations and logical errors. To mitigate this fragility, Self-Consistency (SC) was introduced as a robust decoding strategy (Wang et al., 2022). By sampling multiple diverse reasoning paths and selecting the most consistent final answer (e.g., via majority voting), SC significantly improves answer accuracy and has become a standard technique for enhancing the reliability of LLMs. The intuition is that if an answer appears many times, as the result of different CoT reasoning paths, then it is more likely to be a correct answer instead of a hallucination.

Despite its effectiveness, standard SC relies on a fixed-budget sampling strategy with a pre-determined number of paths for every query to ensure high-confidence results. This redundancy leads to significant latency and operational costs. In response, Adaptive Self-Consistency (ASC) methods have been proposed to improve efficiency by adaptively determining the necessary stopping time, halting generation once a dominant answer emerges (Aggarwal et al., 2023). However, existing ASC frameworks generally overlook prior information regarding the answer distribution's shape, e.g., whether the probability mass is "peaked" (easy query) or "flat" (hard query). Notably, this distribution can be learned based on previous attempts by the same LLM on similar problems. Therefore, the distribution shape (see Lyu et al., 2025) may be effectively estimated from a small subset of historical data, presenting a clear opportunity to enhance ASC efficiency.

Motivated by this idea, we propose an optimal adaptive sampling framework that explicitly integrates prior information, and updates Bayesian posteriors to decide when to stop. Although the conceptual framework is natural, its direct implementation faces a major computational bottleneck. Specifically, calculating the exact posterior probability involves iterating over all possible permutations of answers, leading to a naive computational complexity of $O(K!)$, where $K$ represents the total number of unique answers generated by the LLM for a given query. In open-ended reasoning tasks where $K$ can be large, this computational cost becomes prohibitively expensive, rendering the naive Bayesian update intractable for real-time inference.

Against this backdrop, we now summarize our contributions.

(i) **Bayesian Framework for ASC**. We consider ASC with an informative prior, and derive an optimal Bayesian sampling rule by formulating the stopping decision as a sequential hypothesis testing problem, while combinatorially evaluating the posterior (Section 2).

(ii) **Efficient $L$-Aggregated Posterior Approximation**. To address the factorial complexity of exact posterior computation, we propose a new $L$-aggregated posterior approximation scheme which aggregates the $K$ candidate answers into $L \leq K$ groups, reducing the computational

[1]Stern School of Business, New York University, New York, USA [2]Graduate School of Business, Columbia University, New York, USA. Correspondence to: Zhengyuan Zhou <zz26@stern.nyu.edu>.

*Proceedings of the 43rd International Conference on Machine Learning*, Seoul, South Korea. PMLR 306, 2026. Copyright 2026 by the author(s).

complexity to $O(K^L)$. Theoretically, we prove that our approximated posterior remains unbiased and achieves asymptotic optimality for any $L \geq 3$, that is, the (rate of the) asymptotic stopping time is identical to that under the exact posterior when the confidence level $1 - \delta$ tends to 1.

(iii) **Theoretical Superiority and Empirical Validation**. We provide a rigorous analysis of stopping times under different prior conditions. For known priors: even the coarsest approximation ($L = 2$) yields a smaller (better) asymptotic stopping time compared to prior-free ASC baselines; For uncertain priors where the true prior belongs to one of several candidates: although now $L = 2$ may suffer from efficiency degradation, $L \geq 3$ is still sufficient to outperform ASC baselines. Extensive experiments on real-world datasets validate our theoretical findings, confirming that our approach achieves a significantly smaller stopping time while maintaining high accuracy.

In sum, our $L$-aggregated posterior approximation presents a tradeoff between statistical and computational complexity. A coarser approximation ($L = 2$) allows for faster posterior computation, but may suffer higher sample complexity (i.e., more LLM calls) due to the compressed information state. On the other extreme, the full posterior ($L = K$) causes an intractable posterior computation, but in principle minimizes the sample complexity.

Our work yields the surprising takeaway that "three is all you need": a slightly finer-grained posterior than $L = 2$ is sufficient to achieve asymptotic optimality (see Section 3.3), with little slowdown in posterior computation (see Section 3.2). Moreover, this theoretical result holds up resiliently in (non-asymptotic) empirics: $L = 3$ replicates the optimal sample efficiency of $L = K$ (starting at confidence level $1 - \delta = 0.8$), without replicating its posterior computation time (see Table 1).

For concreteness, our $L = 3$ approximation counts occurrences of: the most-frequent answer, the second-most-frequent answer, and all other answers combined. This provides a minimal way to capture two statistics: (i) the occurrence % for the most-frequent answer, and (ii) its difference with the second-most-frequent answer. Intuitions that these are useful statistics have come up before[1], but here we provide new (theoretical and empirical) evidence that the synthesis of (i)–(ii) is really the "sweet spot" for the tradeoff between statistical and computational complexity.

### 1.1. Related Literature

From an application perspective, our work builds on ASC (Aggarwal et al., 2023), which employs Beta-posterior up-

dates *with an uninformative prior* to terminate sampling once the leading answer meets a confidence threshold. Recent studies have fortified this approach with theoretical guarantees (Huang et al., 2025; Feng et al., 2025) or proposed practical variants, such as lightweight window-based stopping rules (Li et al., 2024), difficulty-aware sampling (Wang et al., 2025; Wan et al., 2025a), and reward-guided cost minimization (Wan et al., 2025b). Although the idea behind (Wang et al., 2025) is similar in spirit, none of these papers formalize *optimal* stopping under an *informative prior*, which is important for having a calibrated mode identification rate while minimizing sampling cost.

We believe this also presents an interesting new problem in sequential hypothesis testing (see Wald, 1992): the optimal Bayesian stopping for mode identification. To elaborate, here the prior is on the probabilities of (observing) the most-frequent, second-most-frequent,... answers, *without knowing what these answers are*, as motivated by the application of LLM answer consistency. Prior works on mode identification (see Shah et al., 2020; Jain et al., 2022) focused on the prior-free setting, presumably because the naive definition of a prior would give away the mode, making the problem vacuous. We show that LLMs present a new form of Bayesian prior for mode identification. Prior to our work, Jain et al. (2022), who established the asymptotic optimality of Beta-stopping rules (with uninformative priors), served as the theoretical backbone for the ASC literature (this connection was also pointed out by Feng et al., 2025). Moreover, the power of our $L = 3$ posterior approximation for our specific problem is also surprising, in relation to the general approaches such as MCMC or normal approximations (see Gelman et al., 1995) that are too computationally expensive for real-time applications and fundamentally ill-suited for discrete data structures, making them inapplicable to our setting.

### 1.2. Organization

The rest of the paper is organized as follows. Section 2 introduces the problem formulation and the optimal Bayesian stopping rule. Section 3 presents the posterior approximation scheme and its theoretical guarantees when the prior information is already known. Then in Section 4, we consider an extension where the prior is only distributionally known and justify how this distribution can be experimentally learned in Section 5. The proofs, algorithm and experiment details are deferred to appendices.

## 2. Optimal Stopping under Prior Information

### 2.1. Model Formulation

For a question, an LLM generates a random answer in $\{a_k\}_{k=1}^K$ each time when sampled, following a probabil-

---

[1]For example: (i), (ii) closely align with "agreement", "first-second-distance" in Lyu et al. (2025), and are conceptually related to "one-versus-rest", "one-versus-one" in Jain et al. (2022).

ity vector $\pi := (p_k)_{k=1}^K$ with $\sum_{k=1}^K p_k = 1$ and indexed so that $p_1 > p_2 \geq \cdots \geq p_K > 0$. The LLM is sampled indefinitely, generating a sequence of answers $(a^{(t)})_{t=1}^\infty$ drawn IID following $\pi$. The objective of ASC is to stop once the most-likely answer $a_1$ can be identified with (high) probability of at least $1 - \delta$ for a given (small) $\delta > 0$.

**Observation Model**  As we do not know the exact answer set of $\{a_k\}_{k=1}^K$, we can only compare whether two observed answers are identical or not. Therefore, after $n$ samples, the entire information available to us is the partition of $\{1, \cdots, n\}$ into groups of identical answers, together with the sizes of these groups. Equivalently, if we list the multiplicities of all distinct observed answers, we obtain a multiset of positive integers that sums to $n$. Formally, let $M(n) \leq K$ denote the number of distinct answers observed among the first $n$ samples, and let $\{n_1, \cdots, n_{M(n)}\} \subset \mathbb{Z}_{>0}$, $\sum_{j=1}^{M(n)} n_j = n$ be the multiset of their occurrence counts for $n_1 \geq \cdots \geq n_{M(n)}$. We compress this multiset further into a count-of-counts representation as:

$$\mathcal{C}_n = \{(v_i, c_i)_{i=1}^q\} \quad \text{with} \quad v_1 > v_2 > \cdots > v_q \geq 1,$$

where $q \leq M(n)$ denotes the number of distinct frequency levels, $c_i \geq 1$ denotes the number of distinct answers that appeared exactly $v_i$ times among the $n$ samples, with the following identities holding:

$$\sum_{j=1}^q c_j = M(n), \qquad \sum_{j=1}^q v_j \cdot c_j = n.$$

*Example* 2.1. Suppose we have sampled the sequence of answers $B, A, B, D$ for a multiple-choice question. Then $n = 4$, $M(n) = 3$, $(n_j)_{j=1}^{M(n)} = (2, 1, 1)$, $q = 2$, and $\mathcal{C}_4 = \{(2, 1), (1, 2)\}$.

**Stopping Rule**  We define event $H_i$ as the hypothesis that the most-likely answer $a_1$ is the answer that has appeared exactly $v_i$ times among the $n$ samples for $i = 1, 2, \cdots, q$.[2] Our goal is to stop at the first time $n$ such that the posterior probability of any hypothesis exceeds $1 - \delta$:

$$n^\star := \inf \left\{ n : \max_{i=1,2,\cdots,q} \mathbb{P}(H_i \mid \mathcal{C}_n) \geq 1 - \delta \right\}, \quad (1)$$

where $\mathbb{P}(H_i \mid \mathcal{C}_n)$ stands for the posterior probability of hypothesis $H_i$ under observation set $\mathcal{C}_n$. It is straightforward to conclude that $\max_i \mathbb{P}(H_i \mid \mathcal{C}_n) = \mathbb{P}(H_1 \mid \mathcal{C}_n)$, i.e., the most-frequently observed answer(s) are most likely to be $a_1$. Thus, we will only focus on $\mathbb{P}(H_1 \mid \mathcal{C}_n)$ in our proceeding analysis.

---

[2] If $c_i > 1$, we can break ties by randomly choosing one answer among the $c_i$ answers with the same frequency $v_i$.

## 2.2. Exact Posterior Calculation

We compute the exact posterior $\mathbb{P}(H_1 \mid \mathcal{C}_n)$ when $\pi$ is known. Note that $\mathbb{P}(H_1 \mid \mathcal{C}_n) = \frac{\mathbb{P}(H_1 \cap \mathcal{C}_n)}{\mathbb{P}(\mathcal{C}_n)}$ and the key is to enumerate all latent answer assignments consistent with the observed count-of-counts $\mathcal{C}_n$.

For the probability of $\mathbb{P}(\mathcal{C}_n)$: Conditioned on observing $M(n)$ distinct answers, there is an unknown injective mapping between the $M(n)$ observed distinct answers and the latent labels in $[K]$. Let $\mathfrak{S}_M$ denote the set of *injective* functions $\psi$ from $[M]$ to $[K]$ (i.e., $\psi(j) \neq \psi(j')$ if $j \neq j'$). For any choice $\psi \in \mathfrak{S}_{M(n)}$, we interpret $\psi(j)$ as the latent label of the $j$-th distinct observed answer (under some arbitrary ordering of the distinct answers). Given the concrete count vector $(n_1, \cdots, n_{M(n)})$ (which is uniquely determined by $\mathcal{C}_n$), the likelihood under $\psi$ equals $\prod_{j=1}^{M(n)} p_{\psi(j)}^{n_j}$.

Given $\mathcal{C}_n$ and the corresponding $(n_1, \cdots, n_{M(n)})$, the number of answer permutations leading to it is

$$\frac{n!}{\prod_{j=1}^{M(n)} n_j!} \cdot \frac{1}{\prod_{j=1}^q c_j!} = \frac{n!}{\prod_{j=1}^q (v_j!)^{c_j}} \cdot \frac{1}{\prod_{j=1}^q c_j!},$$

noting that $\prod_{j=1}^{M(n)} n_j! = \prod_{j=1}^q (v_j!)^{c_j}$, and that we divide by $\prod_{j=1}^q c_j!$ to avoid double-counting when different functions $\psi$ lead to the same answer counts. Therefore, the probability $\mathbb{P}(\mathcal{C}_n)$ can be expressed as:

$$\mathbb{P}(\mathcal{C}_n) = \frac{n!}{\prod_{j=1}^q (v_j!)^{c_j}} \cdot \frac{1}{\prod_{j=1}^q c_j!} \sum_{\psi \in \mathfrak{S}_{M(n)}} \prod_{j=1}^{M(n)} p_{\psi(j)}^{n_j}.$$

Similarly, we have

$$\mathbb{P}(H_1 \cap \mathcal{C}_n)$$

$$= \frac{n!}{\prod_{j=1}^q (v_j!)^{c_j}} \frac{1}{\prod_{j=1}^q c_j!} \sum_{\psi \in \mathfrak{S}_{M(n)} : \psi(1)=1} \prod_{j=1}^{M(n)} p_{\psi(j)}^{n_j}.$$

The constant terms would cancel out and we conclude:

$$\mathbb{P}(H_1 \mid \mathcal{C}_n) = \frac{\sum_{\psi \in \mathfrak{S}_{M(n)} : \psi(1)=1} \prod_{j=1}^{M(n)} p_{\psi(j)}^{n_j}}{\sum_{\psi \in \mathfrak{S}_{M(n)}} \prod_{j=1}^{M(n)} p_{\psi(j)}^{n_j}}. \quad (2)$$

*Remark* 2.2. For the simplest case of $K = 2$, the problem degenerates into a standard sequential hypothesis test between two Bernoulli distributions $\text{Bern}(p_1)$ and $\text{Bern}(1 - p_1)$, which enjoys a simple closed-form stopping rule: if $v_1 - v_2 \geq \lceil \frac{\log((1-\delta)/\delta)}{\log(p_1/(1-p_1))} \rceil$ (Siegmund, 2013), output the most-frequent answer as the final answer; continue sampling otherwise.

However for $K > 2$, computing (2) becomes quickly intractable, with runtime growing as $K!$ (assuming we have observed all the unique answers $M(n) = K$). This motivates us to design more computationally efficient stopping rules, which we describe next.

# 3. Efficient Posterior Approximation Scheme

## 3.1. $L$-Aggregated Posterior Approximation

To address the computational issue for large $K$, we introduce an *L-aggregated observation state* $\mathcal{C}_n^L$ for any $L = 2, 3, \cdots, K$, in which only the top-$(L-1)$ most frequent answers are counted explicitly, while the remaining $K - L + 1$ answers and their frequencies are ignored. To achieve this, we first determine the index $d \in \{1, 2, \cdots, q\}$ such that:

$$\sum_{j=1}^{d-1} c_j < \min\{L-1, M(n)\} \leq \sum_{j=1}^{d} c_j,$$

and let

$$c_j' = \begin{cases} c_j & \text{if } j < d \\ \min\{L-1, M(n)\} - \sum_{i=1}^{d-1} c_i & \text{if } j = d \end{cases}$$

We have $\mathcal{C}_n^L := \{(v_j, c_j')_{j=1}^d\}$.

Example: for $\mathcal{C}_n = \{(10, 1), (3, 2), (2, 1)\}$ with $n = 18$, we have $\mathcal{C}_n^2 = \{(10, 1)\}$, $\mathcal{C}_n^3 = \{(10, 1), (3, 1)\}$, $\mathcal{C}_n^4 = \{(10, 1), (3, 2)\}$. The aggregated posterior $\mathcal{C}_n^3 = \{(10, 1), (3, 1)\}$, e.g., indicates that one answer has appeared 10 times, another answer has appeared 3 times; all other answers have appeared at most 3 times, and combined have appeared $18 - 10 - 3 = 5$ times.

We introduce the shorthand notation $L(n) := \min\{L - 1, M(n)\}$ and define $\bar{n}_{L(n)} := n - \sum_{j=1}^{d} v_j \cdot c_j' = n - \sum_{i=1}^{L(n)} n_i$. Now, we derive:

$$\mathbb{P}(\mathcal{C}_n^L) = \frac{n!}{\bar{n}_{L(n)}! \prod_{j=1}^{d}(v_j!)^{c_j'}} \cdot \frac{1}{\prod_{j=1}^{d} c_j'!} \cdot$$
$$\sum_{\psi \in \mathfrak{S}_{L(n)}} \left( \prod_{j=1}^{L(n)} p_{\psi(j)}^{n_j} \right) \cdot \tilde{S}_\psi$$

with $\tilde{S}_\psi$ aggregating the (likelihood of) "other" answer as follows:

$$\tilde{S}_\psi := \sum_{\mathbf{r}^{-\psi}} w(\mathbf{r}) \cdot \frac{\bar{n}_{L(n)}!}{\prod_{j \in [K] \setminus \psi} r_j!} \cdot \prod_{j \in [K] \setminus \psi} p_j^{r_j}, \quad (3)$$

where $[K] \setminus \psi$ denotes indices of $[K] := \{1, \cdots, K\}$ not in the range of $\psi$, i.e. $|[K] \setminus \psi| = K - L(n)$ if $\psi \in \mathfrak{S}_{L(n)}$, while $\mathbf{r}^{-\psi} := (r_j)_{j \in [K] \setminus \psi}$ denotes an allocation vector, where $r_j \in \{0, \cdots, v_d\}$ and $\sum_{j \in [K] \setminus \psi} r_j = \bar{n}_{L(n)}$, and the weight $w(\mathbf{r}) := \binom{c_d' + m(\mathbf{r})}{c_d'}^{-1}$ for $m(\mathbf{r}) := \#\{j \in [K] \setminus \psi : r_j = v_d\}$ denotes the number of unique answers categorized as "other" that have frequency $v_d$.

*Remark* 3.1. We introduce $w(\mathbf{r})$ to correct for the arbitrary assignment of answers with identical frequency $v_d$ to the tracked set (head) or the aggregated set (tail). Since there are a total of $c_d' + m(\mathbf{r})$ such answers, and the split between head and tail is structurally arbitrary for identical frequencies, $w(\mathbf{r})$ corresponds to the probability of the specific observed assignment of $c_d'$ answers to the head. For example: suppose the answer sequence is $A, A, B, C$ ($n = 4$) and we set $L = 2$. Now we have $C_4^2 = \{(2, 1)\}$ and $v_d = 2$. Consider the tail allocation vectors $\mathbf{r} = (2, 0)$ or $\mathbf{r} = (0, 2)$: note that since $C_4^2$ does not record which tail label attains frequency 2, configurations such as $A, A, B, B$ will be counted twice. The weight $w(\mathbf{r})$ (which equals to $1/2$ in this case) just removes this multiplicity by dividing by the number of indistinguishable assignments.

Based on (3) we can derive expressions for $\mathbb{P}(H_1 \cap \mathcal{C}_n^L)$ and $\mathbb{P}(H_1 \mid \mathcal{C}_n^L)$, which look like those derived earlier for $\mathbb{P}(H_1 \cap \mathcal{C}_n)$ and $\mathbb{P}(H_1 \mid \mathcal{C}_n)$. We can then conclude that

$$\mathbb{P}(H_1 \mid \mathcal{C}_n^L) = \frac{\sum_{\psi \in \mathfrak{S}_{L(n)}: \psi(1)=1} \left( \prod_{j=1}^{L(n)} p_{\psi(j)}^{n_j} \right) \cdot \tilde{S}_\psi}{\sum_{\psi \in \mathfrak{S}_{L(n)}} \left( \prod_{j=1}^{L(n)} p_{\psi(j)}^{n_j} \right) \cdot \tilde{S}_\psi},$$

and we refer to $\mathbb{P}(H_1 \mid \mathcal{C}_n^L)$ as the $L$-aggregated posterior approximation of $\mathbb{P}(H_1 \mid \mathcal{C}_n)$.

The intuition for applying the $L$-aggregation is that the most-frequent observation counts $n_1, \cdots, n_{L-1}$ are the biggest determinants of our confidence in the mode, while the posterior update is much faster when we only condition on the compressed information state $\mathcal{C}_n^L$, as we will illustrate in the later subsections.

## 3.2. Algorithm Design and Computational Issues

Given the $L$-aggregated posterior approximation scheme in the previous subsection, our stopping policy follows a simple posterior-threshold rule: at each time step $n$, we sample an answer, update the counts $\mathcal{C}_n$ and condense it to $\mathcal{C}_n^L$. We compute the approximated posterior $\mathbb{P}(H_1 \mid \mathcal{C}_n^L)$, and stop once it exceeds the target confidence level $1 - \delta$. The procedure is summarized in Algorithm 1 in Appendix C due to space limit. It is worth noting that when the aggregation parameter takes the maximal value of $L = K$, we have $\mathcal{C}_n^L \equiv \mathcal{C}_n$, i.e., the compressed counts recover that of the exact counts. In this case, Algorithm 1 exactly recovers the optimal stopping rule under the true posterior $\mathbb{P}(H_1 \mid \mathcal{C}_n)$.

Next, we analyze the computational issue of Algorithm 1. As can be seen, its computation cost is dominated by evaluating $\mathbb{P}(H_1 \mid \mathcal{C}_n^L)$, with overall complexity coming from two main components: (i) iteration of all injective mappings $\psi \in \mathfrak{S}_{L(n)}$; (ii) calculation of aggregation constant $S_\psi$ for each $\psi \in \mathfrak{S}_{L(n)}$.

We explain how to handle these complexities in the hardest case of $L(n) = L-1$. For (i): it is equivalent to picking and permuting $L-1$ answers out of $K$ answers; thus it has the iteration complexity of $O(\frac{K!}{(K-L+1)!})$; For (ii), we establish combinatorial equivalence with a coefficient of a polynomial of order $\bar{n}_{L(n)}$, which can then be computed using dynamic programming (see Appendix C for more details). We point out that for any $\psi \in \mathfrak{S}_{L-1}$, its computational complexity is of order $O((K-L+1) \cdot \bar{n}_{L(n)}^2)$.

As a result, we can conclude that the total computational complexity of deriving $\mathbb{P}(H_1 \mid \mathcal{C}_n^L)$ is $O(\frac{K! \cdot \bar{n}_{L(n)}^2}{(K-L)!})$, which is dominated by $O(K^L \cdot \bar{n}_{L(n)}^2)$. Its key advantage lies in that: the computational complexity is now exponential in the aggregation parameter $L$ rather than the total number of answers $K$. Since $L$ can be chosen as a small constant (e.g., $L = 3$), the inference still remains highly efficient.

### 3.3. Statistical Guarantees

In this subsection, we analyze the asymptotic sample complexity of our adaptive stopping rules, and also compare them to the sampling rule without prior information.

Our $L$-aggregated approximation can be viewed as a "coarsening" of the Bayesian posterior, where every possible (uncondensed) count set $\mathcal{C}_n$ maps to a unique condensed count set $\mathcal{C}_n^L$. We estimate the probability of $H_1$ conditioned on this coarser count set $\mathcal{C}_n^L$, noting that

$$\mathbb{P}(H_1 \mid \mathcal{C}_n^L) = \mathbb{E}[\mathbb{P}(H_1 \mid \mathcal{C}_n) \mid \mathcal{C}_n^L], \qquad (4)$$

where the expectation is taken w.r.t. all $\mathcal{C}_n$'s that map to $\mathcal{C}_n^L$. As a consequence, the aggregated posterior preserves the correct Bayesian belief on average, while making the posterior much easier to compute; however, it does lose some information by compressing the posterior, which would lead to slower stopping on average. We now show that setting $L = 3$ does not slow down the stopping rate asymptotically.

**Asymptotic Optimality** Recall that the optimal stopping time under our $L$-Aggregated Posterior Approximation Scheme which we denote as $n^{\star,L}$ satisfies:

$$n^{\star,L} := \inf \left\{ n : \mathbb{P}(H_1 \mid \mathcal{C}_n^L) \geq 1-\delta \right\}.$$

Next, we will analyze the expected stopping time $\mathbb{E}[n^{\star,L}]$, with its expectation taken w.r.t. the randomness of the condensed counts $\mathcal{C}_n^L$. In order to make a direct comparison and derive some clean theoretical insights, we follow the routine of Jain et al. (2022) who consider the asymptotic expected stopping time without prior information when $\delta \to 0$. Our main result is the following theorem, which exactly characterizes the asymptotic behavior of expected stopping time among different $L$.

**Theorem 3.2** (Asymptotic Stopping Time under $L$-Aggregated Posterior Approximation Scheme). *The expected stopping time $\mathbb{E}[n^{\star,L}]$ satisfies*

- *For $L = 2$: let $D_{\mathrm{KL}}(p_1 \| p_2)$ denote the KL divergence between $\mathrm{Bern}(p_1)$ and $\mathrm{Bern}(p_2)$. We have*

$$\lim_{\delta \to 0} \frac{\mathbb{E}[n^{\star,2}]}{\log(1/\delta)} = \frac{1}{D_{\mathrm{KL}}(p_1 \| p_2)}. \qquad (5)$$

- *For $L = 3, 4, \cdots, K$: we have*

$$\lim_{\delta \to 0} \frac{\mathbb{E}[n^{\star,L}]}{\log(1/\delta)} = \frac{1}{(p_1 - p_2) \cdot \log \frac{p_1}{p_2}}. \qquad (6)$$

*Remark* 3.3. According to Theorem 3.2, it is surprising to find out that for the simple case of $L = 3$, its asymptotic expected stopping time matches that of the exact posterior with $L = K$. This implies that using $L = 3$ yields an asymptotically optimal stopping rule, even though only the top two most frequent observations are tracked explicitly and the rest are aggregated. In particular, no asymptotic statistical efficiency is lost in this case.

*Remark* 3.4. Shah et al. (2020); Jain et al. (2022) derive a prior-free stopping rule whose stopping time $n^{\star,f}$ satisfies

$$\lim_{\delta \to 0} \frac{\mathbb{E}[n^{\star,f}]}{\log(1/\delta)} = \frac{1}{p_1 \cdot \log \frac{2p_1}{p_1+p_2} + p_2 \cdot \log \frac{2p_2}{p_1+p_2}}, \qquad (7)$$

which they show is tight for prior-free mode identification.

For any answer probabilities satisfying $0 < p_2 < p_1 < 1$ and $p_1 + p_2 < 1$, it can be checked that (7) > (5) > (6). Thus, we have the following relationship as $\delta \to 0$:

$$\mathbb{E}[n^{\star,f}] > \mathbb{E}[n^{\star,2}] > \mathbb{E}[n^{\star,3}] = \cdots = \mathbb{E}[n^{\star,K}] = \mathbb{E}[n^{\star}],$$

which illustrates that Bayesian adaptive sampling achieves a strictly smaller asymptotic stopping time even for very coarse posterior tracking with $L = 2$; and that $L = 3$ is sufficient to achieve the optimal stopping time $n^{\star}$ that is obtained from tracking the exact posterior.

## 4. Adaptive Sampling with Uncertain Prior

In many practical situations in LLM test-time scaling, the exact answer-frequency prior $\pi$ of an LLM on a new question may be unknown. Instead, the model may have been evaluated on a collection of related tasks, each inducing its own empirical answer distribution. This naturally motivates a hierarchical setting in which the prior itself is treated as random and drawn from a hyper-prior.

### 4.1. Problem Setup and Sampling Rule

**Model Formulation** Suppose that the LLM has been evaluated on $M$ related questions, thus producing $M$ empirical answer-frequency distributions

$$\pi^m = (p_{1,m}, p_{2,m}, \cdots, p_{K,m}), \quad m \in [M],$$

where we again index so that $p_{1,m} > p_{2,m} \geq \cdots \geq p_{K,m} \geq 0$ for all $m \in [M]$. To unify the notation, we define $K$ as the maximum number of distinct answers observed among all the $M$ priors. For any task whose answer set contains fewer than $K$ elements, we augment its support to size $K$ by introducing additional "null" answers and assigning them probability 0. This ensures that all priors $\pi^m$ are represented on a common $K$-dimensional simplex.

Let $\Pi^M = \{\pi^1, \cdots, \pi^M\}$ denote the set of candidate priors and $\pi := (p_1, p_2, \cdots, p_K)$ again stand for the true prior for the new question. We assume that $\pi \in \Pi^M$ with the hyper-prior $\lambda_m := \mathbb{P}(\pi = \pi^m)$ with $0 \leq \lambda_m \leq 1$ for all $m \in [M]$ and $\sum_{m=1}^{M} \lambda_m = 1$.

Given historical observations $\mathcal{C}_n$, we can directly follow the routine in (2) to calculate the posterior probability:

$$\mathbb{P}_{\Pi^M}(H_1 \mid \mathcal{C}_n) \tag{8}$$
$$= \frac{\sum_{m=1}^{M} \lambda_m \cdot \sum_{\psi \in \mathfrak{S}_{M(n)}:\psi(1)=1} \prod_{j=1}^{M(n)} p_{\psi(j),m}^{n_j}}{\sum_{m=1}^{M} \lambda_m \cdot \sum_{\psi \in \mathfrak{S}_{M(n)}} \prod_{j=1}^{M(n)} p_{\psi(j),m}^{n_j}},$$

where we use $\mathbb{P}_{\Pi^M}$ to emphasize that it is for the case of uncertain prior within set $\Pi^M$, and the stopping rule is the same as that in (1).

**$L$-Aggregated Posterior Approximation with Uncertain Prior** We extend our $L$-aggregated posterior approximation to the uncertain prior setting. With the condensed historical counts $\mathcal{C}_n^L$ defined as before, we now have:

$$\mathbb{P}_{\Pi^M}(H_1 \mid \mathcal{C}_n^L)$$
$$= \frac{\sum_{m=1}^{M} \lambda_m \cdot \sum_{\psi \in \mathfrak{S}_{L(n)}:\psi(1)=1} \left(\prod_{j=1}^{L(n)} p_{\psi(j),m}^{n_j}\right) \cdot \tilde{S}_\psi^m}{\sum_{m=1}^{M} \lambda_m \cdot \sum_{\psi \in \mathfrak{S}_{L(n)}} \left(\prod_{j=1}^{L(n)} p_{\psi(j),m}^{n_j}\right) \cdot \tilde{S}_\psi^m},$$

where $\tilde{S}_\psi^m$ collects the likelihood contribution of the "other" answers under prior $\pi^m$, defined as:

$$\tilde{S}_\psi^m = \sum_{\mathbf{r}^{-\psi}} w(\mathbf{r}) \cdot \frac{\bar{n}_{L(n)}!}{\prod_{j \in [K] \setminus \psi} r_j!} \cdot \prod_{j \in [K] \setminus \psi} p_{j,m}^{r_j},$$

with the same definitions for $\mathbf{r}^{-\psi}$ and $w(\mathbf{r})$.

The stopping algorithm is identical to before, except with posteriors now evaluated using the hyper-prior-weighted expression (8). This introduces a multiplicative factor of $M$ in computing the posterior, which is insignificant compared

to our $L$-aggregated posterior reducing the posterior computation time from $O(K!)$ to $O(K^L)$. We now consider how many samples it takes the $L$-aggregated posterior to stop.

### 4.2. Performance Guarantees

By the same "coarsening" argument as in (4), our $L$-aggregated posterior under uncertain prior still preserves the correct Bayesian belief. We denote its stopping time using

$$n_{\Pi^M}^{\star,L} := \inf \left\{ n : \mathbb{P}_{\Pi^M}(H_1 \mid \mathcal{C}_n^L) \geq 1 - \delta \right\}.$$

The following Theorem summarizes the asymptotic expected stopping time with uncertain prior.

**Theorem 4.1** (Asymptotic Stopping Time with Uncertain Prior). *The expected stopping time $\mathbb{E}[n_{\Pi^M}^{\star,L}]$ satisfies:*

- *For $L = 2$: define $\rho := \frac{p_1}{1-p_1}$ and $\rho^\dagger := \frac{p_{1,m}}{1-p_{2,m}}$. Let $m^\dagger := \arg\min_{m \in [M]} J_{2,m}(p_1)$, where*

$$J_{2,m}(p_1) := D_{KL}(p_1 \| p_{2,m}) +$$
$$(1 - p_1) \cdot \mathbf{1}\{\rho < \rho^\dagger, \rho < 1\} \cdot D_{KL}(\rho \| \rho^\dagger),$$

*we have:*

$$\lim_{\delta \to 0} \frac{\mathbb{E}[n_{\Pi^M}^{\star,2}]}{\log(1/\delta)} = \frac{1}{J_{2,m^\dagger}(p_1)}.$$

- *For $L = 3, 4, \cdots, K$: define $\rho_L := \frac{p_{L-1}}{1-\sum_{i=1}^{L-1} p_i}$ and $\rho_L^m := \frac{p_{L,m}}{1-\sum_{i=1}^{L-1} p_{i,m}}$. Denote $\mathbf{p}_L := (p_1, \cdots, p_{L-1}, 1 - \sum_{i=1}^{L-1} p_i)$ and $\mathbf{p}_{2,1,3,\cdots,L}^m := (p_{2,m}, p_{1,m}, p_{3,m}, \cdots, p_{L-1,m}, 1 - \sum_{i=1}^{L-1} p_{i,m})$. Let $m^\dagger := \arg\min_{m \in [M]} J_{2,m}^L(\mathbf{p}_L)$, where*

$$J_{2,m}^L(\mathbf{p}_L) := D_{KL}(\mathbf{p}_L \| \mathbf{p}_{2,1,3,\cdots,L}^m) +$$
$$\left(1 - \sum_{i=1}^{L-1} p_i\right) \cdot \mathbf{1}\{\rho_L < \rho_L^m, \rho_L < 1\} \cdot D_{KL}(\rho_L \| \rho_L^m),$$

*we have:*

$$\lim_{\delta \to 0} \frac{\mathbb{E}[n_{\Pi^M}^{\star,L}]}{\log(1/\delta)} = \frac{1}{J_{2,m^\dagger}^L(\mathbf{p}_L)}.$$

We let $m^\star \in [M]$ index the true prior (i.e., $\pi = \pi^{m^\star}$) and make the following remarks.

*Remark* 4.2. For the case $L = 2$: its denominator $J_{2,m^\dagger}(p_1)$ is strictly positive (as KL divergence is always non-negative, and when $D_{KL}(p_1 \| p_{2,m^\dagger}) = 0$, we always have $\rho < \rho^\dagger$). However, we note that $J_{2,m^\dagger}(p_1)$ can be arbitrarily close to zero, which leads to a very large asymptotic rate of stopping time. This may occur when there exists a $p_{2,m}$ for $m \in [M]$ that is rather close to $p_{1,m^\star}$. Consider an example of $M = 2$ with:

$$\pi^1 = (0.51, 0.10, \cdots), \quad \pi^2 = (0.49, 0.48, \cdots),$$

and assume $m^\star = 1$. Now we have $\frac{1}{J_{2,m^\dagger}(p_1)} = 555.22$, even significantly larger than the asymptotic rate of prior-free stopping time in (7) (which is 6.64). Thus, although $L = 2$ remains computationally appealing, it can behave poorly in the uncertain prior setting: the mixture of plausible priors may confound the inference so severely when $L = 2$ that the resulting procedure performs would even worse than the prior-free baseline in terms of asymptotic stopping time. The intuition is that: when $L = 2$, the algorithm only tracks the most frequent observation, and struggles to distinguish whether the current leader is the true winner under $\pi^1$ or $\pi^2$ when $p_{1,1} \approx p_{2,1}$, which forces the stopping rule to demand significantly more samples to confirm the winner.

*Remark* 4.3. For the case $L \geq 3$: unlike the results established in Theorem 3.2 before, now $L = 3$ is not enough for asymptotic optimality. However, we note that in certain favorable situations, e.g., when $m^\dagger = m^\star$, we immediately obtain

$$J_{2,m^\dagger}^L(\mathbf{p}_L) = (p_{1,m^\star} - p_{2,m^\star}) \cdot \log \frac{p_{1,m^\star}}{p_{2,m^\star}},$$

which exactly recovers the known-prior asymptotic rate. Besides, we still have the following relationship as $\delta \to 0$:

$$\mathbb{E}[n^{\star,f}] > \mathbb{E}[n_{\Pi^M}^{\star,3}] \geq \cdots \geq \mathbb{E}[n_{\Pi^M}^{\star,K}] \geq \mathbb{E}[n^\star],$$

with all the equalities holding when $m^\dagger = m^\star$, which guarantees that for any $L \geq 3$, our prior-based stopping rule strictly improves upon the prior-free procedure, and the improvement is monotonically non-decreasing in $L$.

We point out that the first inequality of $\mathbb{E}[n^{\star,f}] > \mathbb{E}[n_{\Pi^M}^{\star,3}]$ can be arbitrarily close to equality in certain unfavorable configurations of the prior set $\Pi^M$, when we have $p_{1,m^\dagger} \approx p_{2,m^\dagger} \approx (p_{1,m^\star} + p_{2,m^\star})/2$. This shows that when the prior set contains a distribution that is extremely hard to distinguish (its top-1 answer), the value of knowing the prior set may become marginal. This highlights a key insight: the prior information is significantly more valuable when the candidate set $\Pi^M$ is small and well-separated, rather than merely inclusive of a large number of possibilities.

# 5. Experiments

## 5.1. Experimental Setup

We evaluate our prior-dependent adaptive sampling rules on FEVAL-TTC given by Rumiantsev et al. (2025), which provides a standardized testbed for test-time compute methods based on cached CoT generations. For each dataset and each LLM, FEVAL-TTC provides pre-recorded CoT responses together with extracted final answers, enabling us to "replay" sequential sampling without issuing live LLM queries. The benchmark covers multiple LLM families (e.g., LLaMA, Qwen, DeepSeek, Mistral, and GPT), which are queried

with 40 independent samples per question under a standardized few-shot CoT prompt, forming the basic evaluation unit for our experiments. We refer readers to Rumiantsev et al. (2025) for details on prompting and answer extraction.

Our experiments are conducted following the routine: (i) we begin with synthetic datasets to benchmark our approximation scheme (with varying aggregation levels $L$) against the exact algorithm and prior-free baselines; (ii) we then evaluate our method on the real-world FEVAL-TTC datasets, covering both the known prior and uncertain prior scenarios. For synthetic datasets we focus fully on mode identification, whereas for FEVAL-TTC we also report answer accuracy[3].

## 5.2. Impact of Aggregation Level under Known Prior

To rigorously quantify the trade-off between the aggregation parameter $L$ and the optimality of the stopping rule, we first conduct a controlled experiment using synthetic data. Our primary objective is to verify that a small $L = 3$ is sufficient to approximate the exact optimal stopping rule with negligible loss in efficiency.

We consider $\pi = (0.5, 0.2, 0.1, 0.1, 0.05, 0.03, 0.01, 0.01)$ with $K = 8$ answers (robustness checks using other priors are conducted in Appendix D). We compare our proposed approximation scheme with aggregation levels $L \in \{2, 3, 4\}$ against the exact stopping rule (i.e., for $L = 8$). Additionally, we include a representative prior-free baseline, denoted as Adaptive Self-Consistency (ASC), which employs the Beta stopping rule in Aggarwal et al. (2023), to demonstrate the efficiency gains obtained by leveraging prior information. We repeat the experiments for 10000 times and evaluate the average performance across varying confidence thresholds $1 - \delta$ based on (i) Mode Estimation Accuracy (Mode Acc.): the fraction of instances whose returned mode matches the true mode of $\pi$; (ii) Number of Generation (Num. Gen.): the number of generations requested before the algorithm terminates. The results are summarized in Table 1, with last column illustrating their average computational time (in milliseconds).

First, comparing different aggregation levels, we observe that $L \geq 3$ yields a stopping rule that is nearly indistinguishable from the exact method for large $1 - \delta$. In contrast, the coarsest aggregation $L = 2$ exhibits a noticeable efficiency penalty, requiring approximately 20% more samples at high confidence levels. These empirical findings are exactly in line with our Theorem 3.2. Besides, we also notice that the average runtime for $L = 3$ (14.2 ms) is a slight increase from that of $L = 2$ (9.0 ms). In contrast, increasing $L$ from 3 to 4 more than doubles the runtime to 37.4 ms, highlighting $L = 3$ as the "sweet spot" for posterior approximation

---

[3]The codes are available through https://github.com/jh9959-afk/Paper.

*Table 1.* Comparison across $L \in \{2, 3, 4\}$ over different thresholds $1 - \delta$.

| Method | $1-\delta = 0.7$ | | $1-\delta = 0.8$ | | $1-\delta = 0.9$ | | $1-\delta = 0.95$ | | $1-\delta = 0.975$ | | $1-\delta = 0.99$ | | |
|---|---|---|---|---|---|---|---|---|---|---|---|---|---|
| | Mode Acc. | Num. Gen. | Mode Acc. | Num. Gen. | Mode Acc. | Num. Gen. | Mode Acc. | Num. Gen. | Mode Acc. | Num. Gen. | Mode Acc. | Num. Gen. | Time |
| $L=2$ | 76.5% | 4.30 | 87.3% | 7.21 | 93.9% | 10.74 | 96.7% | 13.92 | 98.6% | 18.19 | 99.5% | 22.43 | 9.0 |
| $L=3$ | 76.0% | 4.16 | 87.1% | 6.70 | 94.1% | 10.12 | 96.4% | 12.38 | 97.8% | 14.38 | 99.2% | 18.07 | 14.2 |
| $L=4$ | 75.1% | 3.95 | 87.1% | 6.70 | 94.1% | 10.12 | 96.2% | 12.04 | 97.8% | 14.40 | 99.2% | 18.11 | 37.4 |
| Exact | 75.1% | 3.95 | 87.1% | 6.70 | 94.1% | 10.12 | 96.2% | 12.05 | 97.9% | 14.45 | 99.2% | 18.13 | 29.8 |
| ASC | 50.4% | 1.00 | 86.5% | 7.27 | 97.8% | 16.72 | 99.5% | 24.64 | 99.9% | 32.78 | 100.0% | 44.07 | - |

granularity. Note that here the exact stopping rule is even faster than $L = 4$, since $K$ is moderate in this case and the term $\bar{n}_{L(n)}$ dominates.

Second, the advantage of incorporating prior information is evident. The prior-free ASC method reveals a lack of calibration to the specified confidence level $1 - \delta$. As evidenced in Table 1, ASC tends to over-sample at high $1 - \delta$, exceeding the target accuracy at the cost of excessive computation. However, at lower confidence levels, it flips to being overly aggressive, stopping too early and failing to satisfy the target accuracy guarantees.

### 5.3. Real-World Evaluation on FEVAL-TTC

Following the simulation experiments in synthetic datasets, we now evaluate the performance of our framework on real-world datasets sampled in FEVAL-TTC. To evaluate the efficacy of our adaptive sampling rules in a controlled yet realistic environment, we construct the experimental setting as follows.

**Known Prior Construction**   For every question and every LLM, we utilize the raw 40 generations provided in the dataset to compute an empirical answer-frequency distribution. We treat this empirical distribution as the ground-truth distribution $\pi$ for the specific query. In the "known prior" setting, the algorithm has full access to this specific $\pi$.

**Uncertain Prior Set Construction**   In the "uncertain prior" setting, we assume the exact $\pi$ is unknown. To construct the candidate prior set $\Pi^M$, we randomly partition the dataset into a training set (70%) and a testing set (30%). The empirical answer distributions from the training set are collected to form the candidate set $\Pi^M$. During evaluation on the testing set, the algorithm relies solely on this $\Pi^M$ without accessing the ground-truth distribution of the test queries. We further assume a uniform distribution over these candidates, i.e., $\lambda_m \equiv \frac{1}{M}$ for all $m \in [M]$.

To simulate the stochastic nature of LLM generation, we do not merely replay the original sequence. Instead, we generate new, synthetic generation trajectories of length 100 by subsampling (with replacement) from $\pi$.

We evaluate the performance of our framework with $L =$

$2, 3$ by comparing it against the same ASC baseline introduced before. We also record the Answer Accuracy (Ans. Acc., the fraction of questions whose returned answer matches the ground-truth answer). The experiments are conducted under three datasets: CommonsenseQA (for easy commonsense reasoning questions), DisambiguationQA (for hard commonsense reasoning questions), and GSM8K (for arithmetic reasoning) for three LLMs: Qwen-2.5-72B, GPT-4o mini, and LLaMA-3.1-405B. Table 2 presents results under Dataset CommonsenseQA for the three LLM models repeated for 5 times in the testing set and we leave the results for other two datasets to Appendix D, which demonstrate similar results and insights.

The results highlight three key insights driven by the model's high intrinsic consistency (which makes the mode accuracy fairly high and nearly invariant with small $\delta$ for ASC). First, in the *known prior* setting, our method is uniformly more sample-efficient than the ASC baseline while essentially matching its mode accuracy, signifying a Pareto improvement from leveraging the exact prior. Second, in the more practical *uncertain prior* setting, our method is again more sample-efficient than ASC while still satisfying the $1-\delta$ confidence thresholds (although not matching the exact mode accuracy of ASC). Interestingly, when $1 - \delta = 0.95$, the uncertain prior algorithms stop after a single sample, which saves more than 80% in sampling costs compared to ASC while still meeting the mode accuracy threshold of 0.95. Finally, despite these drastic reductions in sampling cost and a slight reduction in mode accuracy, the answer accuracy of our uncertain prior methods remains comparable to that of the conservative ASC baseline and can even be higher than ASC in some scenarios. We note that this phenomenon is not merely coincidental: empirical evidence indicates that the true answer often lies within the Top-2 candidates (with 93.3% accuracy) rather than the mode alone (with 87.6% accuracy) in Dataset CommonsenseQA for these LLMs. By employing early stopping, our method retains a higher probability of selecting the runner-up candidate (i.e., the answer corresponds to $p_2$) than ASC, sometimes recovering the true answer in ambiguous cases where the mode is incorrect, as is also observed in the study of Chen et al. (2024). Understanding exactly when and why the runner-ups outperform the mode remains a compelling subject for future investigation for efficient inference of LLM answers.

*Table 2.* Comparison across methods and models under Dataset CommonsenseQA. We highlight in **bold** the comparison between ASC, the simple prior-free method, and our $L = 3$ (uncertain prior) method, where the uncertain prior can be realistically learned from data. Our method yields a much better tradeoff between Answer Accuracy vs. Number Generated, often with Pareto improvements.

| Model | Method | $1 - \delta = 0.95$ | | | $1 - \delta = 0.975$ | | | $1 - \delta = 0.99$ | | |
|---|---|---|---|---|---|---|---|---|---|---|
| | | Ans. Acc. | Mode Acc. | Num. Gen. | Ans. Acc. | Mode Acc. | Num. Gen. | Ans. Acc. | Mode Acc. | Num. Gen. |
| Qwen-2.5-72B | $L = 2$ (known prior) | 87.5% | 98.9% | 3.41 | 88.0% | 99.4% | 3.74 | 88.0% | 99.4% | 4.27 |
| | $L = 3$ (known prior) | 87.5% | 99.0% | 3.38 | 88.0% | 99.4% | 3.68 | 88.0% | 99.4% | 4.24 |
| | $L = 2$ (uncertain prior) | 86.3% | 95.3% | 1.00 | 87.9% | 98.8% | 4.15 | 88.1% | 99.5% | 6.41 |
| | $L = 3$ (uncertain prior) | **86.3%** | **95.3%** | **1.00** | **87.9%** | **98.8%** | **4.13** | **88.1%** | **99.5%** | **6.23** |
| | ASC | **87.2%** | **99.6%** | **5.28** | **87.6%** | **100.0%** | **6.75** | **87.6%** | **100.0%** | **8.04** |
| GPT-4o mini | $L = 2$ (known prior) | 85.8% | 99.6% | 2.63 | 85.8% | 99.6% | 2.84 | 85.8% | 99.6% | 3.04 |
| | $L = 3$ (known prior) | 85.8% | 99.6% | 2.59 | 85.8% | 99.6% | 2.83 | 85.8% | 99.6% | 3.00 |
| | $L = 2$ (uncertain prior) | 85.5% | 96.9% | 1.00 | 85.5% | 99.1% | 3.43 | 85.8% | 99.6% | 5.39 |
| | $L = 3$ (uncertain prior) | **85.5%** | **96.9%** | **1.00** | **85.5%** | **99.1%** | **3.26** | **85.8%** | **99.6%** | **5.13** |
| | ASC | **86.0%** | **100.0%** | **5.10** | **86.0%** | **100.0%** | **6.31** | **86.0%** | **100.0%** | **7.56** |
| LLaMA-3.1-405B | $L = 2$ (known prior) | 88.8% | 98.5% | 6.92 | 89.3% | 99.0% | 7.72 | 89.4% | 99.0% | 8.71 |
| | $L = 3$ (known prior) | 88.6% | 98.3% | 6.50 | 89.3% | 99.0% | 7.40 | 89.4% | 99.0% | 8.19 |
| | $L = 2$ (uncertain prior) | 88.7% | 97.5% | 5.63 | 89.3% | 98.6% | 8.67 | 89.1% | 98.7% | 10.61 |
| | $L = 3$ (uncertain prior) | **88.6%** | **97.4%** | **4.58** | **89.2%** | **98.5%** | **7.42** | **89.3%** | **98.6%** | **9.10** |
| | ASC | **90.4%** | **99.6%** | **7.55** | **90.0%** | **100.0%** | **9.60** | **90.0%** | **100.0%** | **12.03** |

# 6. Conclusion

In this work, we bridged the gap between Bayesian optimal stopping and LLM test-time scaling. By demonstrating that a coarse, $L$-aggregated posterior ($L = 3$) achieves asymptotic optimality, we established a practical framework that substantially reduces LLM sampling costs without sacrificing the accuracy gains of self-consistency. Overall, our framework provides a theoretically grounded "sweet spot" for balancing computational complexity and statistical efficiency during real-time inference.

While our framework can help achieve significant sampling cost savings for LLM test-time scaling, we acknowledge several practical limitations and open questions: (i) Cold-start issue: Our framework leverages historical data to estimate prior information, thereby improving sampling efficiency. For completely new domains or scenarios without any collected data, it reverts to the standard prior-free ASC baseline. (ii) Prior sensitivity: In cases where the estimated prior is biased, the theoretical performance guarantees on the robustness of our algorithms remain an open question to be explored. (iii) Non-mode ground truth: In scenarios where the LLM's inherent mode answer is not actually the ground truth, how to appropriately extend our framework to address this discrepancy presents an interesting question for future research.

# Acknowledgments

This work is supported by ONR-13983263 and 2027 New York University Center for Global Economy and Business grant.

# Impact Statement

This paper presents work whose goal is to advance the field of Machine Learning, specifically by optimizing the inference efficiency of LLMs. By significantly reducing the number of samples required for accurate reasoning, our method contributes to lowering the operational costs, energy consumption, and computational barriers associated with deploying large models. There are many potential societal consequences of our work, none which we feel must be specifically highlighted here.

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

## A. Appendix: Proof of Main Results

For simplicity, we will treat all $M(n)$ (and therefore $L(n)$) in the paper as $K$ (and $L-1$) in the appendix.

*Proof of Theorem 3.2.* Here we present the main idea in proving the result. We start with the case of $L = 2$ and then generalize the idea to the general $L \geq 3$.

- For the case of $L = 2$: We let $A_i := p_i^{n_1} \cdot S_i$ and $\tilde{A}_i := p_i^{n_1} \cdot \tilde{S}_i$ with $S_i$ introduced as follows for any $i \in [K]$:

$$S_i := \sum_{\mathbf{r}^{(-i)}} \frac{(n-n_1)!}{\prod_{j \neq i} r_j!} \prod_{j \neq i} p_j^{r_j} = (1-p_i)^{n-n_1} \cdot \mathbb{P}^{(-i)}(\mathbf{r}^{(-i)} \in \mathcal{R}_{n,n_1}), \tag{9}$$

where the event $\mathcal{R}_{n,n_1} := \{\mathbf{r}^{(-i)} : \sum_{j \neq i} r_j = n - n_1, \max_{j \neq i} r_j \leq n_1\}$ and $\mathbb{P}^{(-i)}$ denotes the Multinomial distribution such that we have:

$$\mathbf{r}^{(-i)} \sim \text{Mult}\left(n - n_1, \mathbf{q}^{(-i)}\right), \quad \mathbf{q}^{(-i)} := (q_j^{(-i)})_{j \neq i}, \quad q_j^{(-i)} := \frac{p_j}{1 - p_i} \; \forall j \neq i.$$

Note that compared with $\tilde{S}_\psi$ (which we can abbreviate as $\tilde{S}_i$ for the $L = 2$ case), $S_i$ can be interpreted as an approximation of $\tilde{S}_i$ with $w(\mathbf{r}) \equiv 1$. As $w(\mathbf{r})$ in $\tilde{S}_i$ is always no greater than 1, we can conclude $\tilde{S}_i \leq S_i$ for any $i \in [K]$.

Before proceeding, we first introduce the following auxiliary Lemma with its detailed proof left in Appendix B.

**Lemma A.1.** *For the $S_i$ and $\tilde{S}_i$ defined above, we have:*

$$1 \leq \frac{S_{i+1}}{S_i} \leq \left(\frac{p_i}{p_{i+1}}\right)^{n_1}, \qquad 1 \leq \frac{\tilde{S}_{i+1}}{\tilde{S}_i} \leq \left(\frac{p_i}{p_{i+1}}\right)^{n_1}.$$

According to Lemma A.1, we can conclude:

$$\frac{A_{i+1}}{A_i} = \left(\frac{p_{i+1}}{p_i}\right)^{n_1} \cdot \frac{S_{i+1}}{S_i} \leq 1, \qquad \frac{\tilde{A}_{i+1}}{\tilde{A}_i} = \left(\frac{p_{i+1}}{p_i}\right)^{n_1} \cdot \frac{\tilde{S}_{i+1}}{\tilde{S}_i} \leq 1$$

This just illustrates the fact that $A_1 > A_2 \geq \cdots \geq A_K$ and $\tilde{A}_1 > \tilde{A}_2 \geq \cdots \geq \tilde{A}_K$ hold (actually we only require the latter hold, which is enough for the proof). Recall that the optimal stopping rule satisfies $\frac{\tilde{A}_1}{\sum_{i=1}^{K} \tilde{A}_i} \geq 1 - \delta$, which is equivalent to $\frac{\tilde{A}_1}{\sum_{i=2}^{K} \tilde{A}_i} \geq \frac{1-\delta}{\delta}$. We can thus apply the following sandwich bound

$$\frac{\tilde{A}_1}{(K-1) \cdot \tilde{A}_2} \leq \frac{\tilde{A}_1}{\sum_{i=2}^{K} \tilde{A}_i} \leq \frac{\tilde{A}_1}{\tilde{A}_2}.$$

Instead of directly analyzing $\frac{\tilde{A}_1}{\tilde{A}_2}$, we will turn to the simpler $\frac{A_1}{A_2}$. We first present the connection between $\frac{\tilde{A}_1}{\tilde{A}_2}$ and $\frac{A_1}{A_2}$.

For $\frac{A_1}{\tilde{A}_1} = \frac{S_1}{\tilde{S}_1}$: it can be further decomposed as

$$\frac{S_1}{\tilde{S}_1} = \frac{\sum_{\mathbf{r}^{(-1)}:\max_j r_j \leq v_d - 1} \frac{(n-n_1)!}{\prod_{j \neq 1} r_j!} \prod_{j \neq 1} p_j^{r_j} + \sum_{\mathbf{r}^{(-1)}:\max_j r_j = v_d} \frac{(n-n_1)!}{\prod_{j \neq 1} r_j!} \prod_{j \neq 1} p_j^{r_j}}{\sum_{\mathbf{r}^{(-1)}:\max_j r_j \leq v_d - 1} \frac{(n-n_1)!}{\prod_{j \neq 1} r_j!} \prod_{j \neq 1} p_j^{r_j} + \sum_{\mathbf{r}^{(-1)}:\max_j r_j = v_d} w(\mathbf{r}) \cdot \frac{(n-n_1)!}{\prod_{j \neq 1} r_j!} \prod_{j \neq 1} p_j^{r_j}} =: \frac{S_1^{\text{no-tie}} + S_1^{\text{tie}}}{S_1^{\text{no-tie}} + \tilde{S}_1^{\text{tie}}}$$

$$\leq 1 + \frac{S_1^{\text{tie}}}{S_1^{\text{no-tie}}},$$

where $S_1^{\text{no-tie}}$ denotes the total likelihood when $r_j$ does not hit the boundary $v_d$ (i.e., $n_{L-1}$), which takes the same value for both $S_1$ and $\tilde{S}_1$; and $S_1^{\text{tie}}$, $\tilde{S}_1^{\text{tie}}$ denote the total likelihood when there exists $r_j$ hit the boundary $v_d$ (which takes different value for $S_1$ and $\tilde{S}_1$), and the last inequality holds due to $\tilde{S}_1^{\text{tie}} \geq 0$.

Now in both $S_1^{\text{tie}}$ and $S_1^{\text{no-tie}}$ we do not have the complex weight $w(\mathbf{r})$. Next, note that by the definition of $S_1^{\text{tie}}$ and $S_1^{\text{no-tie}}$, we have

$$\frac{S_1^{\text{tie}}}{S_1^{\text{no-tie}}} \leq \frac{\mathbb{P}^{(-1)}(\max_j r_j \geq n_1)}{1 - \mathbb{P}^{(-1)}(\max_j r_j \geq n_1)}.$$

Thus it suffices to upper bound $\mathbb{P}^{(-1)}(\max_j r_j \geq n_1)$. By applying the union bound and that each $r_j \sim \text{Bin}(n-n_1, q_j^{(-1)})$ and the Chernoff bound for Binomial distribution, we have

$$\mathbb{P}^{(-1)}(\max_j r_j \geq n_1) \leq \sum_{j=2}^K \mathbb{P}^{(-1)}(r_j \geq n_1)$$

$$\leq (K-1) \cdot \mathbb{P}\left(\text{Bin}\left(n - n_1, q_2^{(-1)}\right) \geq n_1\right)$$

$$\leq (K-1) \cdot \exp(-c_0 \cdot n)$$

This result holds according to the assumption that $p_1 > p_2$. We omit the detailed analysis here and refer readers to the analysis of (11), which is exactly the same. The main idea is that: we can safely bounded $\mathbb{P}^{(-1)}(\max_j r_j \geq n_1)$ by any constant strictly smaller than 1 for sufficiently large $n$ (as is the case in our asymptotic regime). Thus, we can simply treat $\frac{S_1^{\text{tie}}}{S_1^{\text{no-tie}}} \leq \frac{1/2}{1-1/2} = 1$. We note that the ratio should be a function of $n$ and will converge to 0 exponentially fast, while we can still simply treat it as any positive constant, which will not affect our final results. The same analysis also applies for $\frac{S_2}{\tilde{S}_2}$.

Therefore, we have

$$\frac{A_1}{2(K-1) \cdot A_2} \leq \frac{\tilde{A}_1}{(K-1) \cdot \tilde{A}_2} \leq \frac{\tilde{A}_1}{\sum_{i=2}^K \tilde{A}_i} \leq \frac{\tilde{A}_1}{\tilde{A}_2} \leq \frac{2A_1}{A_2}.$$

Define:

$$n_{\text{low}} := \inf\left\{n : \log\frac{A_1}{A_2} \geq \log\frac{1-\delta}{\delta} - \log 2\right\}, \quad n_{\text{upp}} := \inf\left\{n : \log\frac{A_1}{A_2} \geq \log\frac{1-\delta}{\delta} + \log(K-1) + \log 2\right\},$$

where $n_{\text{low}}$ corresponds to the stopping time when $\log\frac{A_1}{A_2} \geq \log\frac{1-\delta}{\delta} - \log 2$ and $n_{\text{upp}}$ corresponds to the stopping time when $\log\frac{A_1}{2(K-1) \cdot A_2} \geq \log\frac{1-\delta}{\delta}$. Following the sandwich bound, we have $n_{\text{low}} \leq n^{\star,2} \leq n_{\text{upp}}$ holds a.s.

We use $Z_n$ to denote the log-likelihood ratio by period $n$ (i.e., sampling for $n$ times) that $Z_n := \log\frac{A_1}{A_2} = \sum_{t=1}^n Y_t$. Now, by the equivalent form of $S_i = (1-p_i)^{n-n_1} \cdot \mathbb{P}^{(-i)}(\mathbf{r} \in \mathcal{R}_{n,n_1})$, $Y_t$ can be defined through:

$$Y_{t+1} = \begin{cases} \log\frac{p_1}{p_2} + \log\frac{\mathbb{P}^{(-1)}(\mathbf{r} \in \mathcal{R}_{t+1,t_1+1})}{\mathbb{P}^{(-1)}(\mathbf{r} \in \mathcal{R}_{t,t_1})} - \log\frac{\mathbb{P}^{(-2)}(\mathbf{r} \in \mathcal{R}_{t+1,t_1+1})}{\mathbb{P}^{(-2)}(\mathbf{r} \in \mathcal{R}_{t,t_1})}, & \text{Observe } u_1 \text{ at period } t+1 \\ \log\frac{1-p_1}{1-p_2} + \log\frac{\mathbb{P}^{(-1)}(\mathbf{r} \in \mathcal{R}_{t+1,t_1})}{\mathbb{P}^{(-1)}(\mathbf{r} \in \mathcal{R}_{t,t_1})} - \log\frac{\mathbb{P}^{(-2)}(\mathbf{r} \in \mathcal{R}_{t+1,t_1})}{\mathbb{P}^{(-2)}(\mathbf{r} \in \mathcal{R}_{t,t_1})}, & \text{Observe others} \end{cases}$$

where we use $t_1$ to denote the number of most frequent observations by period $t$. We further define $\tilde{Z}_n := n_1 \cdot \log\frac{p_1}{p_2} + (n-n_1) \cdot \log\frac{1-p_1}{1-p_2}$ and $\Lambda_n := \log\mathbb{P}^{(-1)}(\mathbf{r} \in \mathcal{R}_{n,n_1}) - \log\mathbb{P}^{(-2)}(\mathbf{r} \in \mathcal{R}_{n,n_1})$. Thus, we can decompose $Z_n$ into $Z_n = \tilde{Z}_n + \Lambda_n$. Define $\phi(x) := x\log\frac{p_1}{p_2} + (1-x)\log\frac{1-p_1}{1-p_2}$ and let $J_t \in \{1, 2, \cdots, K\}$ denote the "true type" of the most frequent observation denoted as $u_1$ (in the dataset $\mathcal{C}_t^2$). We can also introduce $\tilde{Y}_t$ as follows:

$$\tilde{Y}_{t+1} = \begin{cases} \log\frac{p_1}{p_2}, & \text{Observe } u_1 \text{ at period } t+1 \\ \log\frac{1-p_1}{1-p_2}, & \text{Observe others} \end{cases}$$

Accordingly, we have $Y_{t+1} = \tilde{Y}_{t+1} + \Delta_{t+1}^{(-1)} - \Delta_{t+1}^{(-2)}$, where $\Delta_{t+1}^{(-i)} := \log \frac{\mathbb{P}^{(-i)}(\mathbf{r} \in \mathcal{R}_{t+1,t_1+\mathbf{1}\{o_{t+1}=u_1\}})}{\mathbb{P}^{(-i)}(\mathbf{r} \in \mathcal{R}_{t,t_1})}$ for any $i = 1, 2$ and we use $\mathbf{1}\{o_{t+1} = u_1\} = 1$ to denote observing $u_1$ at period $t + 1$ and $\mathbf{1}\{o_{t+1} = u_1\} = 0$ to denote observation answers other than $u_1$.

We use $\mu_t := \mathbb{E}[Y_{t+1} \mid \mathcal{C}_t^2]$ to denote the conditional single-period drift at period $t$ given historical observation $\mathcal{C}_t^2$, which exists since the single-step increment $|Y_{t+1}|$ is bounded by a constant almost surely. The expected single-period drift at period $t$ can be expressed as:

$$\mathbb{E}[\mu_t] = \mathbb{E}[Y_{t+1}] = \sum_{i=1}^{K} \phi(p_i) \cdot \mathbb{P}(J_t = i) + \epsilon_t, \qquad \epsilon_t := \mathbb{E}[\Delta_{t+1}^{(-1)} - \Delta_{t+1}^{(-2)}].$$

Define $\Delta_i := p_1 - p_i > 0$ and we use $t_i := \sum_{\ell=1}^{t} \mathbf{1}\{o_\ell = a_i\}$ to denote the number of observations of answer $i$ by period $t$. For any $i = 2, 3, \cdots, K$, according to Hoeffding's inequality,

$$\mathbb{P}(J_t = i) \leq \mathbb{P}(t_i \geq t_1) = \mathbb{P}\left(\sum_{\ell=1}^{t}(\mathbf{1}\{o_\ell = a_i\} - \mathbf{1}\{o_\ell = a_1\}) \geq 0\right)$$

$$= \mathbb{P}\left(\sum_{\ell=1}^{t}(\mathbf{1}\{o_\ell = a_i\} - \mathbf{1}\{o_\ell = a_1\}) + t\Delta_i \geq t\Delta_i\right)$$

$$\leq \exp(-2\Delta_i^2 t).$$

As a result,

$$\mathbb{P}(J_t \neq 1) \leq \sum_{i=2}^{K} \exp(-2\Delta_i^2 t) \leq (K-1) \cdot \exp(-2\Delta_1^2 t). \tag{10}$$

Next, we will bound the deviation $\epsilon_t$. For any $\eta \in (0, \min\{\frac{p_1-p_2}{4}, p_2\})$, define random event $\mathcal{G}_t := \{|t_1 - tp_1| \leq t\eta\}$. Again, by Hoeffding's inequality, $\mathbb{P}(\mathcal{G}_t^c) \leq 2\exp(-2\eta^2 t)$. Under $\mathcal{G}_t$, we have

$$\frac{t_1}{t - t_1} \in [\alpha_l, \alpha_u], \qquad \alpha_l := \frac{p_1 - \eta}{1 - p_1 + \eta}, \qquad \alpha_u := \frac{p_1 + \eta}{1 - p_1 - \eta},$$

and for any $s \in \{0, 1\}$, $\frac{t_1+s}{t-t_1} \geq \alpha_l$ by definition. According to the definition of $\mathcal{R}_{t,t_1}$, we have its complement set $\mathcal{R}_{t,t_1}^c = \{\exists j : r_j \geq t_1 + 1\}$. For any $i \in \{1, 2\}$ and $j \neq i$, under the probability measure $\mathbb{P}^{(-i)}$, we have $r_j \sim \text{Bin}(t - t_1, q_j^{(-i)})$ (recall that $q_j^{(-i)} := \frac{p_j}{1-p_i}$). Thus, for any $s \in \{0, 1\}$, by the Chernoff bound for Binomial distribution,

$$\mathbb{P}^{(-i)}(r_j \geq t_1 + s) = \mathbb{P}^{(-i)}\left(\frac{r_j}{t - t_1} \geq \frac{t_1 + s}{t - t_1}\right) \leq \exp\left(-(t - t_1) \cdot D_{\text{KL}}(\theta_{t,s} \| q_j^{(-i)})\right), \tag{11}$$

with $\theta_{t,s} := \frac{t_1+s}{t-t_1} \geq \alpha_l$. Next, we will show that there exists a constant $c_1 > 0$ such that $\alpha_l - q_j^{(-i)} \geq c_1$ holds for any $i \in \{1, 2\}$ and $j \neq i$:

(1) When $i = 1$: $q_j^{(-1)} \leq q_2^{(-1)} = p_2/(1 - p_1)$. By the fact that $p_1 > p_2$ and $\eta < (p_1 - p_2)/4$, we have:

$$\alpha_l - q_j^{(-1)} \geq \frac{p_1 - p_2}{2(1 - p_1 + \eta)} := c_{1,1} > 0.$$

(2) When $i = 2$: $q_j^{(-2)} \leq q_1^{(-2)} = p_1/(1 - p_2)$. Similarly, we have:

$$\alpha_l - q_j^{(-2)} \geq \frac{p_1 \cdot (p_1 - p_2)}{2(1 - p_1 + \eta) \cdot (1 - p_2)} := c_{1,2} > 0.$$

By choosing $c_1 = \min\{c_{1,1}, c_{1,2}\}$, we have $\alpha_l - q_j^{(-i)} \geq c_1$. The KL divergence $D_{KL}(\theta_{t,s} || q_j^{(-i)})$ can therefore be lower bounded by a constant $c_2 > 0$. Thus, under the random event $\mathcal{G}_t$ (specifically, under $t - t_1 \geq (1 - p_1 - \eta) \cdot t$), let the constant $c_3 := c_2 \cdot (1 - p_1 - \eta)$, we conclude that by the union bound,

$$1 - \mathbb{P}^{(-i)}(\mathbf{r} \in \mathcal{R}_{t,t_1}) = \mathbb{P}^{(-i)}(\mathbf{r} \in \mathcal{R}_{t,t_1}^c) \leq (K - 1) \cdot \exp(-c_3 \cdot t),$$

and so does

$$1 - \mathbb{P}^{(-i)}(\mathbf{r} \in \mathcal{R}_{t+1,t_1+s}) \leq (K - 1) \cdot \exp(-c_3 \cdot t)$$

holds for any $i \in \{1, 2\}$ and $s \in \{0, 1\}$. Therefore, there exists a $T_0 > 0$ such that when $t > T_0$, $(K-1) \cdot \exp(-c_3 t) \leq 1/2$. According to the inequality that $|\log(1 - x)| \leq 2x$ for any $x \in [0, 1/2]$, and under the event $\mathcal{G}_t$, we have:

$$\left| \log \frac{\mathbb{P}^{(-i)}(\mathbf{r} \in \mathcal{R}_{t+1,t_1+s})}{\mathbb{P}^{(-i)}(\mathbf{r} \in \mathcal{R}_{t,t_1})} \right| \leq |\log \mathbb{P}^{(-i)}(\mathbf{r} \in \mathcal{R}_{t+1,t_1+s})| + |\log \mathbb{P}^{(-i)}(\mathbf{r} \in \mathcal{R}_{t,t_1})|$$

$$= 2 \cdot (1 - \mathbb{P}(\mathbf{r} \in \mathcal{R}_{t+1,t_1+s})) + 2 \cdot (1 - \mathbb{P}(\mathbf{r} \in \mathcal{R}_{t,t_1}))$$

$$\leq 4(K - 1) \cdot \exp(-c_3 t)$$

holds for any $i \in \{1, 2\}$ and $s \in \{0, 1\}$. Then we apply the following decomposition of the expectation and obtain:

$$\mathbb{E}[|\Delta_{t+1}^{(-i)}|] = \mathbb{E}[|\Delta_{t+1}^{(-i)}| \mid \mathcal{G}_t] + \mathbb{E}[|\Delta_{t+1}^{(-i)}| \mid \mathcal{G}_t^c] \leq c_4(K - 1)\exp(-c_5 t)$$

for some constants $c_4, c_5 > 0$ given that $|\Delta_{t+1}^{(-i)}|$ is upper bounded by a constant. As a result, $|\epsilon_t| \leq 2c_4(K-1)\exp(-c_5 t)$. Consequently, we have:

$$|\mathbb{E}[\mu_t] - \phi(p_1)| \leq c_6(K - 1)\exp(-c_7 t)$$

for some constants $c_6, c_7 > 0$ for any $t \geq T_0$. At last, we have the expected log-likelihood ratio at the stopping time $n_{\text{low}}$:

$$\mathbb{E}[Z_{n_{\text{low}}}] = \mathbb{E}\left[\sum_{t \geq 0} Y_{t+1} \cdot \mathbf{1}\{t < n_{\text{low}}\}\right] = \sum_{t \geq 0} \mathbb{E}[Y_{t+1} \cdot \mathbf{1}\{t < n_{\text{low}}\}] = \sum_{t \geq 0} \mathbb{E}[\mathbb{E}[Y_{t+1} \cdot \mathbf{1}\{t < n_{\text{low}}\} \mid \mathcal{O}_t^2]]$$

$$= \sum_{t \geq 0} \mathbb{E}[\mathbb{E}[Y_{t+1} \cdot \mathbf{1}\{t < n_{\text{low}}\} \mid \mathcal{O}_t^2]] = \sum_{t \geq 0} \mathbb{E}[\mathbf{1}\{t < n_{\text{low}}\} \cdot \mathbb{E}[Y_{t+1} \mid \mathcal{O}_t^2]]$$

$$= \sum_{t \geq 0} \mathbb{E}[\mathbf{1}\{t < n_{\text{low}}\} \cdot \mu_t] = \sum_{t \geq 0} \mathbb{E}[\mathbf{1}\{t < n_{\text{low}}\} \cdot (\phi(p_1) + (\mu_t - \phi(p_1)))]$$

$$= \phi(p_1) \cdot \mathbb{E}[n_{\text{low}}] + \sum_{t \geq 0} \mathbb{E}[(\mu_t - \phi(p_1)) \cdot \mathbf{1}\{t < n_{\text{low}}\}].$$

with $\sum_{t \geq 0} \mathbb{E}|(\mu_t - \phi(p_1)) \cdot \mathbf{1}\{t < n_{\text{low}}\}| \leq c_8 K$ for some constant $c_8 > 0$. Following the definition of $n_{\text{low}}$, we thus have $\log(1 - \delta)/\delta - \log 2 \leq \mathbb{E}[Z_{n_{\text{low}}}] \leq \log(1 - \delta)/\delta - \log 2 + c_9$ (for some constant $0 \leq c_9 < 1$ due to the rounding issue). As a result, we conclude:

$$1 - \frac{\log 2}{\log(1 - \delta)/\delta} \leq \frac{\phi(p_1) \cdot \mathbb{E}[n_{\text{low}}]}{\log(1 - \delta)/\delta} + \frac{\sum_{t \geq 0} \mathbb{E}[(\mu_t - \phi(p_1)) \cdot \mathbf{1}\{t < n_{\text{low}}\}]}{\log(1 - \delta)/\delta} \leq 1 - \frac{2 - c_9}{\log(1 - \delta)/\delta}.$$

Similarly, we also have for some constant $0 \leq c_{10} < 1$:

$$1 + \frac{2 + \log(K - 1)}{\log(1 - \delta)/\delta} \leq \frac{\phi(p_1) \cdot \mathbb{E}[n_{\text{upp}}]}{\log(1 - \delta)/\delta} + \frac{\sum_{t \geq 0} \mathbb{E}[(\mu_t - \phi(p_1)) \cdot \mathbf{1}\{t < n_{\text{upp}}\}]}{\log(1 - \delta)/\delta} \leq 1 + \frac{2 + c_{10} + \log(K - 1)}{\log(1 - \delta)/\delta}.$$

Thus, by the fact that $n_{\text{low}} \leq n^{\star,2} \leq n_{\text{upp}}$ holds a.s., we have:

$$\lim_{\delta \to 0} \frac{\mathbb{E}[n^{\star,2}]}{\log(1/\delta)} = \frac{1}{D_{KL}(p_1 || p_2)}.$$

($\delta$ should be the order of $o(\exp(-K))$.)

- For the case of $L = 3, \cdots, K$: Similarly, we introduce $A_{i_1,i_2,\cdots,i_{L-1}} := S_{i_1,i_2,\cdots,i_{L-1}} \cdot \prod_{t=1}^{L-1} p_{i_t}^{n_t}$ and $\tilde{A}_{i_1,i_2,\cdots,i_{L-1}} :=$
$\tilde{S}_{i_1,i_2,\cdots,i_{L-1}} \cdot \prod_{t=1}^{L-1} p_{i_t}^{n_t}$, with $S_{i_1,i_2,\cdots,i_{L-1}}$ defined as follows:

$$S_{i_1,i_2,\cdots,i_{L-1}} := (1 - \sum_{t=1}^{L-1} p_{i_t})^{n - \sum_{t=1}^{L-1} n_t} \cdot \mathbb{P}^{-(i_1,\cdots,i_{L-1})}(\mathbf{r} \in \mathcal{R}_{n,n_1,\cdots,n_{L-1}}),$$

where $\mathcal{R}_{n,n_1,\cdots,n_{L-1}} := \{\mathbf{r} : \sum_{j \neq i_1,\cdots,i_{L-1}} r_j = n - \sum_{t=1}^{L-1} n_t, \max_{j \neq i_1,\cdots,i_{L-1}} r_j \leq n_{L-1}\}$. Below we also introduce another auxiliary Lemma with its detailed proof left in Appendix B.

**Lemma A.2.** *For the $A_{i_1,i_2,\cdots,i_{L-1}}$ defined above, we have:*

$$\max_{i_2,\cdots,i_{L-1}} A_{1,i_2,\cdots,i_{L-1}} = A_{1,2,\cdots,L-1}, \quad \max_{i_1 \neq 1} A_{i_1,i_2,\cdots,i_{L-1}} = A_{2,1,3,\cdots,L-1}; \tag{12}$$

$$\max_{i_2,\cdots,i_{L-1}} \tilde{A}_{1,i_2,\cdots,i_{L-1}} = \tilde{A}_{1,2,\cdots,L-1}, \quad \max_{i_1 \neq 1} \tilde{A}_{i_1,i_2,\cdots,i_{L-1}} = \tilde{A}_{2,1,3,\cdots,L-1}; \tag{13}$$

Recall that now the stopping rule satisfies $\frac{\sum_{\psi:\psi(1)=1} \tilde{A}_\psi}{\sum_\psi \tilde{A}_\psi} \geq 1 - \delta$. By applying Lemma A.2 and the same analysis as before, we can turn it into the following sandwich inequalities:

$$\frac{(K - L + 1)! \cdot A_{1,2,3,\cdots,L-1}}{2(K-1) \cdot (K-1)! \cdot A_{2,1,3,\cdots,L-1}} \leq \frac{\sum_{\psi:\psi(1)=1} \tilde{A}_\psi}{\sum_{\psi:\psi(1)\neq 1} \tilde{A}_\psi} \leq \frac{2(K-1)! \cdot A_{1,2,3,\cdots,L-1}}{(K - L + 1)! \cdot A_{2,1,3,\cdots,L-1}}.$$

Similarly, now we can define $n_{\text{low}}$ and $n_{\text{upp}}$ through (we throw away the $\log 2$ constants for simplicity reason, which will not affect our final conclusions):

$$n_{\text{low}} := \inf \left\{ n : \log \frac{A_{1,2,3,\cdots,L-1}}{A_{2,1,3,\cdots,L-1}} \geq \log \frac{1-\delta}{\delta} - \log(K-1)! + \log(K - L + 1)! \right\}$$

$$n_{\text{upp}} := \inf \left\{ n : \log \frac{A_{1,2,3,\cdots,L-1}}{A_{2,1,3,\cdots,L-1}} \geq \log \frac{1-\delta}{\delta} + \log(K-1) + \log(K-1)! - \log(K - L + 1)! \right\},$$

and following the sandwich bound, we still have $n_{\text{low}} \leq n^{\star,L} \leq n_{\text{upp}}$ holds a.s.

Thus, it suffices to focus on the term of $\frac{A_{1,2,3,\cdots,L-1}}{A_{2,1,3,\cdots,L-1}}$. We can define the log-likelihood ratio through

$$Z_n := \log \frac{A_{1,2,3\cdots,L-1}}{A_{2,1,3,\cdots,L-1}} = (n_1 - n_2) \cdot \log \frac{p_1}{p_2} := \sum_{t=1}^{n} Y_t,$$

where the equality holds due to the fact that $S_{1,2,3\cdots,L-1} = S_{2,1,3,\cdots,L-1}$. As a result, now $Y_t$ has a simpler expression as that of the $L = 2$ case:

$$Y_{t+1} = \begin{cases} \log \dfrac{p_1}{p_2}, & \text{Observe } u_1 \text{ at period } t+1 \\ \log \dfrac{p_2}{p_1}, & \text{Observe } u_2 \text{ at period } t+1 \\ 0, & \text{Observe other answers besides } u_1 \text{ and } u_2 \text{ at period } t+1. \end{cases}$$

Again, we use $\mu_t := \mathbb{E}[Y_{t+1} \mid \mathcal{O}_t^L]$ to denote the conditional single-period drift at period $t$ given historical observation $\mathcal{O}_t^L$; we use $J_t^{(1)} \in [K]$ and $J_t^{(2)} \in [K]$ to denote the "true type" of $u_1$ and $u_2$ at period $t$, respectively. Let $\phi(p_i, p_j) := (p_i - p_j) \cdot \log \frac{p_1}{p_2}$. The expected single-period drift at period $t$ can now be expressed as:

$$\mathbb{E}[\mu_t] = \mathbb{E}[Y_{t+1}] = \sum_{i \neq j} \phi(p_i, p_j) \cdot \mathbb{P}(J_t^{(1)} = i, J_t^{(2)} = j),$$

and we want to bound the term of $|\mathbb{E}[\mu_t] - \phi(p_1, p_2)|$. Define set $\mathcal{S}_2 := \{j \in [K] : p_j = p_2\}$, and let $m := |\mathcal{S}_2|$. We can thus define "good event" as $\mathcal{E}_t := \{J_t^{(1)} = 1\} \cap \{J_t^{(2)} \in \mathcal{S}_2\}$. When the good event $\mathcal{E}_t$ holds, we would have $\mathbb{E}[\mu_t] = \phi(p_1, p_2)$. Besides, since $Y_{t+1} \in \left\{\log \dfrac{p_1}{p_2}, \log \dfrac{p_2}{p_1}, 0\right\}$, we always have $|Y_{t+1}| \leq \log \dfrac{p_1}{p_2}$. Therefore,

$$|\mathbb{E}[\mu_t] - \phi(p_1, p_2)| = |\mathbb{E}[(\mu_t - \phi(p_1, p_2)) \cdot \mathbf{1}\{\mathcal{E}_t^c\}]| \leq 2\log \frac{p_1}{p_2} \cdot \mathbb{P}(\mathcal{E}_t^c).$$

Next, we will upper bound the term of $\mathbb{P}(\mathcal{E}_t^c)$. Note that $\mathbb{P}(\mathcal{E}_t^c) \leq \mathbb{P}(J_t^{(1)} \neq 1) + \mathbb{P}(J_t^{(2)} \notin \mathcal{S}_2)$, and we will bound the two terms separately. Recall that $\Delta_i = p_1 - p_i > 0$. We can directly follow the analysis in (10) and again conclude:

$$\mathbb{P}(J_t^{(1)} \neq 1) \leq (K - 1) \cdot \exp(-2\Delta_1^2 t). \tag{14}$$

For the term $\mathbb{P}(J_t^{(2)} \notin \mathcal{S}_2)$: as $\mathbb{P}(J_t^{(2)} \notin \mathcal{S}_2) = \mathbb{P}(J_t^{(1)} = 1, J_t^{(2)} \notin \mathcal{S}_2) + \mathbb{P}(J_t^{(1)} \neq 1, J_t^{(2)} \notin \mathcal{S}_2) \leq \mathbb{P}(J_t^{(1)} = 1, J_t^{(2)} \notin \mathcal{S}_2) + \mathbb{P}(J_t^{(1)} \neq 1)$, it suffices to bound $\mathbb{P}(J_t^{(1)} = 1, J_t^{(2)} \notin \mathcal{S}_2)$. Consider two cases: (i) the trivial case where $\mathcal{S}_2 = \{2, 3, \cdots, K\}$, i.e., $p_2 = p_3 = \cdots = p_K$. Now $\mathbb{P}(J_t^{(1)} = 1, J_t^{(2)} \notin \mathcal{S}_2) = 0$; (ii) $\mathcal{S}_2 = \{2, \cdots, m+1\}$ where $m \leq K - 2$, i.e., $p_2 = \cdots = p_{m+1} > p_{m+2}$, and we can define the (weak) gap of $\Gamma := p_2 - p_{m+2} > 0$. Recall that $t_i = \sum_{\ell=1}^t \mathbf{1}\{o_\ell = a_i\}$ is the number of observations (of answer) $i$ by period $t$. In this case, we take any fixed $j^* \in \mathcal{S}_2$, and for any $i \notin \{1\} \cup \mathcal{S}_2$. By applying the Hoeffding's inequality,

$$\mathbb{P}(t_i \geq t_{j^*}) = \mathbb{P}\left(\sum_{\ell=1}^t (\mathbf{1}\{o_\ell = a_i\} - \mathbf{1}\{o_\ell = a_{j^*}\}) \geq 0\right) \leq \exp(-2\Gamma^2 t).$$

Thus, by the fact that $\{J_t^{(1)} = 1, J_t^{(2)} \notin \mathcal{S}_2\} \subseteq \cup_{i \notin \{1\} \cup \mathcal{S}_2}\{t_i \geq t_{j^*}\}$ and applying the union bound,

$$\mathbb{P}(J_t^{(1)} = 1, J_t^{(2)} \notin \mathcal{S}_2) \leq \sum_{i \notin \{1\} \cup \mathcal{S}_2} \mathbb{P}(t_i \geq t_{j^*}) \leq (K - 1 - m) \cdot \exp(-2\Gamma^2 t). \tag{15}$$

Combining (14) and (15) together, we can obtain:

$$\mathbb{P}(\mathcal{E}_t^c) \leq 2(K - 1) \cdot \exp(-2\Delta_1^2 t) + (K - 1 - m) \cdot \exp(-2\Gamma^2 t) \leq c_{11} \cdot \exp(-c_{12} \cdot t)$$

for some constants $c_{11}, c_{12} > 0$. As a result,

$$|\mathbb{E}[\mu_t] - \phi(p_1, p_2)| \leq 2c_{11} \log \frac{p_1}{p_2} \cdot \exp(-c_{13} \cdot t) =: c_{13} \cdot \exp(-c_{12} \cdot t)$$

for some constant $c_{13} > 0$ (independent of $\delta$). The remaining analysis directly follows that in the case of $L = 2$ and we can get:

$$\mathbb{E}[Z_{n_{\text{low}}}] = \phi(p_1, p_2) \cdot \mathbb{E}[n_{\text{low}}] + \sum_{t \geq 0} \mathbb{E}\left[(\mu_t - \phi(p_1, p_2)) \cdot \mathbf{1}\{t < n_{\text{low}}\}\right],$$

and similar for $\mathbb{E}[Z_{n_{\text{upp}}}]$. By the same analysis, we can conclude:

$$\lim_{\delta \to 0} \frac{\mathbb{E}[n^{\star, L}]}{\log(1/\delta)} = \frac{1}{(p_1 - p_2) \cdot \log \dfrac{p_1}{p_2}}.$$

($\delta$ should be of the order $o(\exp(-K^{1+\epsilon}))$ for any $\epsilon > 0$.)

$\square$

*Proof of Theorem 4.1.* We start with the case of $L = 2$ and then generalize the idea to $L \geq 3$ as well.

- For the case of $L = 2$: Denote $A_i^m := p_{i,m}^{n_1} \cdot S_i^m$ and $\tilde{A}_i^m := p_{i,m}^{n_1} \cdot \tilde{S}_i^m$, where $S_i^m$ just follows the definition of $S_i$ introduced in the proof of Theorem 3.2. Similar as before, we need to figure out the largest $\tilde{A}_i^m$ among all $i \in [K]$. Note that for any $m$, we still have $\tilde{A}_i^m \geq \tilde{A}_{i+1}^m$ holds according to the analysis in Lemma A.1. Thus, we can conclude $\max_{i,m} \tilde{A}_i^m = \max_m \tilde{A}_1^m$. Besides, for each $m \in [M]$, we have $\sum_{i=2}^K \tilde{A}_i^m \in [\tilde{A}_2^m, (K-1) \cdot \tilde{A}_2^m]$. We further define $\hat{m}^\star := \operatorname{argmax}_{m \in [M]} \tilde{A}_1^m$ and $\hat{m}^\dagger := \operatorname{argmax}_{m \in [M]} \tilde{A}_2^m$, and $\lambda_{min} := \min_{m \in [M]} \lambda_m$.

We can again apply the sandwich bound and obtain:

$$\frac{\lambda_{min}}{K-1} \cdot \frac{\tilde{A}_1^{\hat{m}^\star}}{\tilde{A}_2^{\hat{m}^\dagger}} \leq \frac{\sum_{m=1}^M \lambda_m \tilde{A}_1^m}{\sum_{m=1}^M \lambda_m \sum_{i=2}^K \tilde{A}_i^m} \leq \frac{K-1}{\lambda_{min}} \cdot \frac{\tilde{A}_1^{\hat{m}^\star}}{\tilde{A}_2^{\hat{m}^\dagger}}.$$

To determine the $\hat{m}^\star$ and $\hat{m}^\dagger$, we first introduce the shorthand notations $\rho := \dfrac{p_1}{1-p_1}$, $\rho_{1,m}^{(-2)} := \dfrac{p_{1,m}}{1-p_{2,m}}$, $\rho_{2,m}^{(-1)} := \dfrac{p_{2,m}}{1-p_{1,m}}$ and the following auxiliary Lemma with its detailed proof left in Appendix B.

**Lemma A.3.** *Let* $B_1^m := \lim_{n \to \infty} \frac{1}{n} \log A_1^m$, $B_2^m := \lim_{n \to \infty} \frac{1}{n} \log A_2^m$ *and* $\tilde{B}_1^m := \lim_{n \to \infty} \frac{1}{n} \log \tilde{A}_1^m$, $\tilde{B}_2^m := \lim_{n \to \infty} \frac{1}{n} \log \tilde{A}_2^m$, *we have:*

$$B_1^m = \tilde{B}_1^m = p_1 \cdot \log p_{1,m} + (1-p_1) \cdot \log(1-p_{1,m}) - (1-p_1) \cdot D_{\mathrm{KL}}(\rho \| \rho_{2,m}^{(-1)}) \cdot \mathbf{1}\{\rho < \rho_{2,m}^{(-1)}\}, \tag{16}$$

*and*

$$B_2^m = \tilde{B}_2^m = p_1 \cdot \log p_{2,m} + (1-p_1) \cdot \log(1-p_{2,m}) - (1-p_1) \cdot D_{\mathrm{KL}}(\rho \| \rho_{1,m}^{(-2)}) \cdot \mathbf{1}\{\rho < \rho_{1,m}^{(-2)}\}. \tag{17}$$

According to Lemma A.3, we define:

$$J_{1,m}(p_1) := D_{\mathrm{KL}}(p_1 \| p_{1,m}) + (1-p_1) \cdot \mathbf{1}\{\rho < \rho_{2,m}^{(-1)}\} \cdot D_{\mathrm{KL}}(\rho \| \rho_{2,m}^{(-1)}),$$
$$J_{2,m}(p_1) := D_{\mathrm{KL}}(p_1 \| p_{2,m}) + (1-p_1) \cdot \mathbf{1}\{\rho < \rho_{1,m}^{(-2)}\} \cdot D_{\mathrm{KL}}(\rho \| \rho_{1,m}^{(-2)}),$$

and define the minimizers $m^\star \in \operatorname{argmin}_{m \in [M]} J_{1,m}(p_1)$ and $m^\dagger \in \operatorname{argmin}_{m \in [M]} J_{2,m}(p_1)$. For simplicity, we assume each minimizer is unique. Since $[M]$ is finite, this immediately implies positive gaps:

$$\Delta^\star := \min_{m \neq m^\star} [J_{1,m}(p_1) - J_{1,m^\star}(p_1)] > 0, \quad \Delta^\dagger := \min_{m \neq m^\dagger} [J_{2,m}(p_1) - J_{2,m^\dagger}(p_1)] > 0.$$

We note that once the true prior $\pi$ indeed belongs to one of the priors in the candidate set $\Pi^M$, we would always have $J_{1,m^\star}(p_1) = 0$, while our analysis presented here also holds for the general case of $J_{1,m^\star}(p_1) > 0$ where the true prior does not belong to any of the priors in $\Pi^M$.

By Lemma A.3, $\frac{1}{n} \log \tilde{A}_1^m \to -J_{1,m}(p_1) + p_1 \log p_1 + (1-p_1) \log(1-p_1)$ as $n \to \infty$, where the additive term is independent of $m$. Hence $\hat{m}^\star \neq m^\star$ only if there exists $m \neq m^\star$ such that $\frac{1}{n} \log(\tilde{A}_1^m / \tilde{A}_1^{m^\star}) \geq 0$, an event whose probability decays as $\exp(-c\Delta^\star n)$ for some constant $c > 0$ by standard large-deviation arguments for empirical multinomial counts (see also the analysis leading to (10) and the Sanov-type bound in Lemma A.3). Applying the union bound over $m \in [M]$ and noting that $M$ is finite gives $\mathbb{P}(\hat{m}^\star \neq m^\star) \leq M \exp(-c\Delta^\star n)$, and similarly $\mathbb{P}(\hat{m}^\dagger \neq m^\dagger) \leq M \exp(-c\Delta^\dagger n)$. Both probabilities are negligible relative to the $\log(1/\delta)$ scale of the stopping time. Thus, we can safely ignore the cases of $\hat{m}^\star \neq m^\star$ and $\hat{m}^\dagger \neq m^\dagger$ and replace $\dfrac{\tilde{A}_1^{\hat{m}^\star}}{\tilde{A}_2^{\hat{m}^\dagger}}$ by $\dfrac{\tilde{A}_1^{m^\star}}{\tilde{A}_2^{m^\dagger}}$ in the subsequent analysis. Following the previous analysis on the comparison between $\tilde{A}_1^{m^\star}, A_1^{m^\star}$ and $\tilde{A}_2^{m^\dagger}, A_2^{m^\dagger}$ (note that the $m$ index is fixed in the comparison, so everything remains the same), we can again consider $\dfrac{A_1^{m^\star}}{A_2^{m^\dagger}}$ instead of $\dfrac{\tilde{A}_1^{m^\star}}{\tilde{A}_2^{m^\dagger}}$.

We let $Z_t := \log \dfrac{A_1^{m^\star}}{A_2^{m^\dagger}}$ and define: (ignore the $\log 2$ constant)

$$n_{\mathrm{low}} := \inf \left\{ n : Z_t \geq \log \frac{1-\delta}{\delta} - \log \frac{K-1}{\lambda_{min}} \right\}, \qquad n_{\mathrm{upp}} := \inf \left\{ n : Z_t \geq \log \frac{1-\delta}{\delta} + \log \frac{K-1}{\lambda_{min}} \right\}$$

and the optimal stopping time $n_{\Pi_M}^{\star,2}$ satisfies: $n_{\text{low}} \leq n_{\Pi_M}^{\star,2} \leq n_{\text{upp}}$.

We again use $Y_{t+1} := Z_{t+1} - Z_t$ to denote the single-period drift and the expected single-period drift $\mu_t := \mathbb{E}[Y_{t+1} \mid \mathcal{O}_t^2]$, with its expression: (we use $u_1$ to denote the most frequent observation)

$$
Y_{t+1} = \begin{cases} \log \dfrac{p_{1,m^\star}}{p_{2,m^\dagger}} + \log \dfrac{\mathbb{P}_{m^\star}^{(-1)}(\mathbf{r} \in \mathcal{R}_{t+1,t_1+1})}{\mathbb{P}_{m^\star}^{(-1)}(\mathbf{r} \in \mathcal{R}_{t,t_1})} - \log \dfrac{\mathbb{P}_{m^\dagger}^{(-2)}(\mathbf{r} \in \mathcal{R}_{t+1,t_1+1})}{\mathbb{P}_{m^\dagger}^{(-2)}(\mathbf{r} \in \mathcal{R}_{t,t_1})}, & \text{Observe } u_1 \text{ at period } t+1 \\[2ex] \log \dfrac{1-p_{1,m^\star}}{1-p_{2,m^\dagger}} + \log \dfrac{\mathbb{P}_{m^\star}^{(-1)}(\mathbf{r} \in \mathcal{R}_{t+1,t_1})}{\mathbb{P}_{m^\star}^{(-1)}(\mathbf{r} \in \mathcal{R}_{t,t_1})} - \log \dfrac{\mathbb{P}_{m^\dagger}^{(-2)}(\mathbf{r} \in \mathcal{R}_{t+1,t_1})}{\mathbb{P}_{m^\dagger}^{(-2)}(\mathbf{r} \in \mathcal{R}_{t,t_1})}, & \text{Observe others.} \end{cases}
$$

We slightly abuse the notations here and again define $\Delta_{t+1}^{(i)}(s)$ for any $i \in \{1,2\}$ and $s := \mathbf{1}\{o_{t+1} = u_1\}$ to be $\Delta_{t+1}^{(1)}(s) := \log \frac{\mathbb{P}_{m^\star}^{(-1)}(\mathbf{r} \in \mathcal{R}_{t+1,t_1+s})}{\mathbb{P}_{m^\star}^{(-1)}(\mathbf{r} \in \mathcal{R}_{t,t_1})}$ and $\Delta_{t+1}^{(2)}(s) := \log \frac{\mathbb{P}_{m^\dagger}^{(-2)}(\mathbf{r} \in \mathcal{R}_{t+1,t_1+s})}{\mathbb{P}_{m^\dagger}^{(-2)}(\mathbf{r} \in \mathcal{R}_{t,t_1})}$ and we will next focus on upper bound the terms $\Delta_{t+1}^{(i)}(s)$. Before proceeding, we first introduce the following useful results which are derived following the proof of Lemma A.3:

$$
\lim_{(t-t_1)\to\infty} -\frac{1}{t-t_1} \log \mathbb{P}_{m^\star}^{(-1)}(\mathbf{r} \in \mathcal{R}_{t,t_1}) = D_{\text{KL}}(\rho||\rho_{2,m^\star}^{(-1)}) \cdot \mathbf{1}\{\rho < \rho_{2,m^\star}^{(-1)}\},
$$

$$
\lim_{(t-t_1)\to\infty} -\frac{1}{t-t_1} \log \mathbb{P}_{m^\dagger}^{(-2)}(\mathbf{r} \in \mathcal{R}_{t,t_1}) = D_{\text{KL}}(\rho||\rho_{1,m^\dagger}^{(-2)}) \cdot \mathbf{1}\{\rho < \rho_{1,m^\dagger}^{(-2)}\}.
$$

If $\rho \geq \rho_{2,m^\star}^{(-1)}$, the analysis is trivial. We consider the case of $\rho < \rho_{2,m^\star}^{(-1)}$. Define the (random) ratio parameter $\theta_t := \frac{t_1}{t-t_1}$, we have

$$
\begin{aligned}
\lim_{(t-t_1)\to\infty} \Delta_{t+1}^{(1)}(1) &= \lim_{(t-t_1)\to\infty} \log \frac{\mathbb{P}_{m^\star}^{(-1)}(\mathbf{r} \in \mathcal{R}_{t+1,t_1+1})}{\mathbb{P}_{m^\star}^{(-1)}(\mathbf{r} \in \mathcal{R}_{t,t_1})} = \lim_{(t-t_1)\to\infty} \log \frac{\mathbb{P}_{m^\star}^{(-1)}(\max_{j\neq 1} r_j \leq t_1 + 1, \sum_j r_j = t - t_1)}{\mathbb{P}_{m^\star}^{(-1)}(\max_{j\neq 1} r_j \leq t_1, \sum_j r_j = t - t_1)} \\
&= \lim_{(t-t_1)\to\infty} -(t-t_1) \cdot \left( D_{\text{KL}}\left( \theta_t + \frac{1}{t-t_1} \Big|\Big| \rho_{2,m^\star}^{(-1)} \right) - D_{\text{KL}}\left( \theta_t || \rho_{2,m^\star}^{(-1)} \right) \right) \\
&= -\frac{\partial}{\partial\theta} D_{\text{KL}}(\theta||\rho_{2,m^\star}^{(-1)}) \Big|_{\theta=\rho} = \log \frac{\rho_{2,m^\star}^{(-1)}(1-\rho)}{\rho(1-\rho_{2,m^\star}^{(-1)})}.
\end{aligned}
$$

Similarly, we have

$$
\begin{aligned}
\lim_{(t-t_1)\to\infty} \Delta_{t+1}^{(1)}(0) &= \lim_{(t-t_1)\to\infty} \log \frac{\mathbb{P}_{m^\star}^{(-1)}(\mathbf{r} \in \mathcal{R}_{t+1,t_1})}{\mathbb{P}_{m^\star}^{(-1)}(\mathbf{r} \in \mathcal{R}_{t,t_1})} = \lim_{(t-t_1)\to\infty} \log \frac{\mathbb{P}_{m^\star}^{(-1)}(\max_{j\neq 1} r_j \leq t_1, \sum_j r_j = t + 1 - t_1)}{\mathbb{P}_{m^\star}^{(-1)}(\max_{j\neq 1} r_j \leq t_1, \sum_j r_j = t - t_1)} \\
&= \lim_{(t-t_1)\to\infty} -(t+1-t_1) \cdot \left( D_{\text{KL}}\left( \theta_t - \frac{\theta_t}{t+1-t_1} \Big|\Big| \rho_{2,m^\star}^{(-1)} \right) - D_{\text{KL}}\left( \theta_t || \rho_{2,m^\star}^{(-1)} \right) \right) \\
&= -\rho \cdot \frac{\partial}{\partial\theta} D_{\text{KL}}(\theta||\rho_{2,m^\star}^{(-1)}) \Big|_{\theta=\rho} - D_{\text{KL}}(\rho||\rho_{2,m^\star}^{(-1)}) = \log \frac{1 - \rho_{2,m^\star}^{(-1)}}{1 - \rho}.
\end{aligned}
$$

And symmetric results also hold for $\Delta_{t+1}^{(2)}(1)$ and $\Delta_{t+1}^{(2)}(0)$. Besides, we have for any $i = 1,2$ and $s = 0,1$, under the "good" event $\mathcal{G}_t := \{|t_1/t - p_1| \leq \eta\}$ for any fixed $\eta \in (0, p_1)$, there exists constant $c_1 > 0$ such that:

$$
\left| \Delta_{t+1}^{(i)}(s) - \lim_{(t-t_1)\to\infty} \Delta_{t+1}^{(i)}(s) \right| \leq \frac{c_1}{t - t_1}
$$

due to the (local) Lipschitz continuous property of the KL divergence (under $\mathcal{G}_t$) and $|\theta_{t+1} - \theta_t| = O(1/(t-t_1))$.

The corresponding proxy increment $\tilde{Y}_{t+1}$ is now defined through

$$
\tilde{Y}_{t+1} = \begin{cases} \log \dfrac{p_{1,m^\star}}{p_{2,m^\dagger}} + \mathbf{1}\{\rho < \rho_{2,m^\star}^{(-1)}\} \cdot \log \dfrac{\rho_{2,m^\star}^{(-1)}(1-\rho_{2,m^\star}^{(-1)})}{\rho(1-\rho_{2,m^\star}^{(-1)})} - \mathbf{1}\{\rho < \rho_{1,m^\dagger}^{(-2)}\} \cdot \log \dfrac{\rho_{1,m^\dagger}^{(-2)}(1-\rho)}{\rho(1-\rho_{1,m^\dagger}^{(-2)})}, & \text{Observe } u_1 \text{ at period } t+1 \\[2ex] \log \dfrac{1-p_{1,m^\star}}{1-p_{2,m^\dagger}} + \mathbf{1}\{\rho < \rho_{2,m^\star}^{(-1)}\} \cdot \log \dfrac{1-\rho_{2,m^\star}^{(-1)}}{1-\rho} - \mathbf{1}\{\rho < \rho_{1,m^\dagger}^{(-2)}\} \cdot \log \dfrac{1-\rho_{1,m^\dagger}^{(-2)}}{1-\rho}, & \text{Observe others.} \end{cases}
$$

Then, under $\mathcal{G}_t$, we have $|Y_{t+1} - \tilde{Y}_{t+1}| \le c_2 \cdot \frac{1}{t}$ for some constants $c_2 > 0$. By straightforward calculations and following the analysis in (10) (which is omitted here for simplicity), we have for some constants $c_3, c_4 > 0$,

$$|\mathbb{E}[\mu_t] - (J_{2,m^\dagger}(p_1) - J_{1,m^\star}(p_1))| \le \frac{c_2}{t} + c_3(K-1)\exp(-c_4 t). \tag{18}$$

By the key result in (18), we can therefore refer to the previous stopping time analysis and conclude for some constants $c_5 > 0$:

$$1 \le \frac{(J_{2,m^\dagger}(p_1) - J_{1,m^\star}(p_1)) \cdot \mathbb{E}[n_{\text{low}}]}{\log(1-\delta)/\delta} + \frac{\sum_{t \ge 0} \mathbb{E}\left[(\mu_t - (J_{2,m^\dagger}(p_1) - J_{1,m^\star}(p_1))) \cdot \mathbf{1}\{t < n_{\text{low}}\}\right]}{\log(1-\delta)/\delta} \le 1 + \frac{c_5}{\log(1-\delta)/\delta}.$$

As $\mathbb{E}[\sum_{t=1}^{n_{\text{low}}} 1/t] = \mathbb{E}[\log n_{\text{low}}]$ is dominated by $\mathbb{E}[n_{\text{low}}]$, and applying the analysis to $n_{\text{low}}$ as well, we conclude

$$\lim_{\delta \to 0} \frac{\mathbb{E}[n_{\Pi^{[M]}}^{\star,2}]}{\log(1/\delta)} = \frac{1}{J_{2,m^\dagger}(p_1) - J_{1,m^\star}(p_1)} = \frac{1}{J_{2,m^\dagger}(p_1)}.$$

- For the case of $L = 3, \cdots, K$: We will only present the different parts compared to the previous analysis for simplicity. Denote $A_{i_1,i_2,\cdots,i_{L-1}}^m := S_{i_1,i_2,\cdots,i_{L-1}}^m \cdot \prod_{t=1}^{L-1} p_{i_t,m}^{n_t}$, where $S_{i_1,i_2,\cdots,i_{L-1}}^m$ is further defined through:

$$S_{i_1,i_2,\cdots,i_{L-1}}^m := (1 - \sum_{t=1}^{L-1} p_{i_t,m})^{n - \sum_{t=1}^{L-1} n_t} \cdot \mathbb{P}_m^{-(i_1,\cdots,i_{L-1})}(\mathbf{r} \in \mathcal{R}_{n,n_1,\cdots,n_{L-1}}),$$

where $\mathcal{R}_{n,n_1,\cdots,n_{L-1}} := \{\mathbf{r} : \sum_{j \ne i_1,\cdots,i_{L-1}} r_j = n - \sum_{t=1}^{L-1} n_t, \max_{j \ne i_1,\cdots,i_{L-1}} r_j \le n_{L-1}\}$. Now we still have $\max_{m \in [M]} \max_{i_2,\cdots,i_{L-1}} A_{i_1,i_2,\cdots,i_{L-1}}^m = \max_{m \in [M]} A_{1,2,\cdots,L-1}^m$ and $\max_{m \in [M]} \max_{i_1 \ne 1} A_{i_1,i_2,\cdots,i_{L-1}}^m = \max_{m \in [M]} A_{2,1,3,\cdots,L-1}^m$. Now we have

$$B_{1,2,\cdots,L-1}^m := \lim_{n \to \infty} \frac{1}{n} \log A_{1,2,\cdots,L-1}^m$$
$$= \sum_{i=1}^{L-1} p_i \cdot \log p_{i,m} + (1 - \sum_{i=1}^{L-1} p_i) \cdot \log(1 - \sum_{i=1}^{L-1} p_{i,m}) - (1 - \sum_{i=1}^{L-1} p_i) \cdot D_{\text{KL}}(\rho_L \| \rho_L^m) \cdot \mathbf{1}\{\rho_L < \rho_L^m\}, \tag{19}$$

$$B_{2,1,3,\cdots,L-1}^m := \lim_{n \to \infty} \frac{1}{n} \log A_{2,1,3,\cdots,L-1}^m = p_1 \cdot \log p_{2,m} + p_2 \cdot \log p_{1,m} +$$
$$\sum_{i=3}^{L-1} p_i \cdot \log p_{i,m} + (1 - \sum_{i=1}^{L-1} p_i) \cdot \log(1 - \sum_{i=1}^{L-1} p_{i,m}) - (1 - \sum_{i=1}^{L-1} p_i) \cdot D_{\text{KL}}(\rho_L \| \rho_L^m) \cdot \mathbf{1}\{\rho_L < \rho_L^m\}, \tag{20}$$

following the same analysis in Lemma A.3, where we define $\rho_L := \frac{p_{L-1}}{1 - \sum_{i=1}^{L-1} p_i}$ and $\rho_L^m := \frac{p_{L,m}}{1 - \sum_{i=1}^{L-1} p_{i,m}}$. Similarly, we let $m^\star := \text{argmin}_{m \in [M]} B_{1,2,\cdots,L-1}^m$ and $m^\dagger := \text{argmin}_{m \in [M]} B_{2,1,3,\cdots,L-1}^m$. Denote $\mathbf{p}_L := (p_1, \cdots, p_{L-1}, 1 - \sum_{i=1}^{L-1} p_i)$, $\mathbf{p}_{1,2,\cdots,L}^m := (p_{1,m}, \cdots, p_{L-1,m}, 1 - \sum_{i=1}^{L-1} p_{i,m})$ and $\mathbf{p}_{2,1,3,\cdots,L}^m := (p_{2,m}, p_{1,m}, p_{3,m}, \cdots, p_{L-1,m}, 1 - \sum_{i=1}^{L-1} p_{i,m})$. We can conclude that:

$$m^\star := \text{argmin}_{m \in [M]} J_{1,m}^L(\mathbf{p}_L) := \text{argmin}_{m \in [M]} D_{\text{KL}}(\mathbf{p}_L \| \mathbf{p}_{1,2,\cdots,L}^m) + (1 - \sum_{i=1}^{L-1} p_i) \cdot D_{\text{KL}}(\rho_L \| \rho_L^m) \cdot \mathbf{1}\{\rho_L < \rho_L^m\},$$

and

$$m^\dagger := \text{argmin}_{m \in [M]} J_{2,m}^L(\mathbf{p}_L) := \text{argmin}_{m \in [M]} D_{\text{KL}}(\mathbf{p}_L \| \mathbf{p}_{2,1,3\cdots,L}^m) + (1 - \sum_{i=1}^{L-1} p_i) \cdot D_{\text{KL}}(\rho_L \| \rho_L^m) \cdot \mathbf{1}\{\rho_L < \rho_L^m\}.$$

Then, we define the log-likelihood ratio through

$$Z_t := \log \frac{A_{1,2,3\cdots,L-1}^{m^\star}}{A_{2,1,3,\cdots,L-1}^{m^\dagger}},$$

and $Y_{t+1} := Z_{t+1} - Z_t$, where we set (we use $u_i$ to denote the $i$-th frequent observation)

$$
Y_{t+1} = \begin{cases}
\log \dfrac{p_{1,m^\star}}{p_{2,m^\dagger}}, & \text{Obs. } u_1 \\[2ex]
\log \dfrac{p_{2,m^\star}}{p_{1,m^\dagger}}, & \text{Obs. } u_2 \\[2ex]
\log \dfrac{p_{i,m^\star}}{p_{i,m^\dagger}}, & \text{Obs. } u_i, \quad \forall i = 3, \cdots, L-2 \\[2ex]
\log \dfrac{p_{L-1,m^\star}}{p_{L-1,m^\dagger}} + \log \dfrac{\mathbb{P}_{m^\star}^{-(1,\cdots,L-1)}(\mathbf{r} \in \mathcal{R}_{t+1,t_1,\cdots,t_{L-1}+1})}{\mathbb{P}_{m^\star}^{-(1,\cdots,L-1)}(\mathbf{r} \in \mathcal{R}_{t,t_1,\cdots,t_{L-1}})} - \log \dfrac{\mathbb{P}_{m^\dagger}^{-(1,\cdots,L-1)}(\mathbf{r} \in \mathcal{R}_{t+1,t_1,\cdots,t_{L-1}+1})}{\mathbb{P}_{m^\dagger}^{-(1,\cdots,L-1)}(\mathbf{r} \in \mathcal{R}_{t,t_1,\cdots,t_{L-1}})}, & \text{Obs. } u_{L-1} \\[3ex]
\log \dfrac{1 - \sum_{i=1}^{L-1} p_{i,m^\star}}{1 - \sum_{i=1}^{L-1} p_{i,m^\dagger}} + \log \dfrac{\mathbb{P}_{m^\star}^{-(1,\cdots,L-1)}(\mathbf{r} \in \mathcal{R}_{t+1,t_1,\cdots,t_{L-1}})}{\mathbb{P}_{m^\star}^{-(1,\cdots,L-1)}(\mathbf{r} \in \mathcal{R}_{t,t_1,\cdots,t_{L-1}})} - \log \dfrac{\mathbb{P}_{m^\dagger}^{-(1,\cdots,L-1)}(\mathbf{r} \in \mathcal{R}_{t+1,t_1,\cdots,t_{L-1}})}{\mathbb{P}_{m^\dagger}^{-(1,\cdots,L-1)}(\mathbf{r} \in \mathcal{R}_{t,t_1,\cdots,t_{L-1}})}, & \text{Obs. others}
\end{cases}
$$

Similarly, we can argue that for $\theta_t := \dfrac{t_{L-1}}{t - \sum_{i=1}^{L-1} t_i}$, when $\rho_L < \rho_L^{m^\star}$ and we observe $u_{L-1}$ at period $t+1$:

$$
\begin{aligned}
&\lim_{(t - \sum_{i=1}^{L-1} t_i) \to \infty} \log \frac{\mathbb{P}_{m^\star}^{-(1,\cdots,L-1)}(\mathbf{r} \in \mathcal{R}_{t+1,t_1,\cdots,t_{L-1}+1})}{\mathbb{P}_{m^\star}^{-(1,\cdots,L-1)}(\mathbf{r} \in \mathcal{R}_{t,t_1,\cdots,t_{L-1}})} \\
&= \lim_{(t - \sum_{i=1}^{L-1} t_i) \to \infty} \log \frac{\mathbb{P}_{m^\star}^{-(1,\cdots,L-1)}(\max_{j \neq 1,\cdots,L-1} r_j \leq t_{L-1}+1, \sum_{j \neq 1,\cdots,L-1} r_j = t - \sum_{i=1}^{L-1} t_i)}{\mathbb{P}_{m^\star}^{-(1,\cdots,L-1)}(\max_{j \neq 1,\cdots,L-1} r_j \leq t_{L-1}, \sum_{j \neq 1,\cdots,L-1} r_j = t - \sum_{i=1}^{L-1} t_i))} \\
&= \lim_{(t - \sum_{i=1}^{L-1} t_i) \to \infty} -(t - \sum_{i=1}^{L-1} t_i) \cdot \left( D_{\text{KL}} \left( \theta_t + \frac{1}{t - \sum_{i=1}^{L-1} t_i} \middle\| \rho_L^{m^\star} \right) - D_{\text{KL}} \left( \theta_t \| \rho_L^{m^\star} \right) \right) \\
&= \log \frac{\rho_L^{m^\star}(1 - \rho_L)}{\rho_L(1 - \rho_L^{m^\star})}.
\end{aligned}
$$

By combining these results together, we again introduce $\tilde{Y}_{t+1}$ as follows:

$$
\tilde{Y}_{t+1} = \begin{cases}
\log \dfrac{p_{1,m^\star}}{p_{2,m^\dagger}}, & \text{Obs. } u_1 \\[2ex]
\log \dfrac{p_{2,m^\star}}{p_{1,m^\dagger}}, & \text{Obs. } u_2 \\[2ex]
\log \dfrac{p_{i,m^\star}}{p_{i,m^\dagger}}, & \text{Obs. } u_i, \quad \forall i = 3, \cdots, L-2 \\[2ex]
\log \dfrac{p_{L-1,m^\star}}{p_{L-1,m^\dagger}} + \mathbf{1}\{\rho_L < \rho_L^{m^\star}\} \cdot \log \dfrac{\rho_L^{m^\star}(1 - \rho_L)}{\rho_L(1 - \rho_L^{m^\star})} - \mathbf{1}\{\rho_L < \rho_L^{m^\dagger}\} \cdot \log \dfrac{\rho_L^{m^\dagger}(1 - \rho_L)}{\rho_L(1 - \rho_L^{m^\dagger})}, & \text{Obs. } u_{L-1} \\[3ex]
\log \dfrac{1 - \sum_{i=1}^{L-1} p_{i,m^\star}}{1 - \sum_{i=1}^{L-1} p_{i,m^\dagger}} + \mathbf{1}\{\rho_L < \rho_L^{m^\star}\} \cdot \log \dfrac{1 - \rho_L^{m^\star}}{1 - \rho_L} - \mathbf{1}\{\rho_L < \rho_L^{m^\dagger}\} \cdot \log \dfrac{1 - \rho_L^{m^\dagger}}{1 - \rho_L}. & \text{Obs. others}
\end{cases}
$$

We omit the similar analysis routine and finally conclude

$$
\lim_{\delta \to 0} \frac{\mathbb{E}[n_{\Pi^M}^{\star,L}]}{\log(1/\delta)} = \frac{1}{J_{2,m^\dagger}^L(\mathbf{p}_L) - J_{1,m^\star}^L(\mathbf{p}_L)} = \frac{1}{J_{2,m^\dagger}^L(\mathbf{p}_L)}.
$$

$\square$

## B. Appendix: Proof of Auxiliary Results

*Proof of Lemma A.1.* Denote $u := n_1$. For any $0 \le r \le u$, we have:

$$p_i^r \le \left(\frac{p_i}{p_{i+1}}\right)^u \cdot p_{i+1}^r, \qquad p_i^r \ge p_{i+1}^r.$$

Recall that $S_i$ has an equivalent form of $S_i = (n - n_1)! \, [z^{n-n_1}] \left(\prod_{j \ne i} \sum_{r=0}^{\min\{n_1, n-n_1\}} \frac{(p_j z)^r}{r!}\right)$ where we use $[z^n]$ to

denote the coefficient of $z^n$. Then we let $T_u(z) := \sum_{r=0}^{u} \frac{z^r}{r!}$ and have:

$$T_u(p_i z) \le_{\text{coef}} \left(\frac{p_i}{p_{i+1}}\right)^u T_u(p_{i+1} z), \quad T_u(p_i z) \ge_{\text{coef}} T_u(p_{i+1} z),$$

based on the non-increasing property of $p_i$, where we use $\le_{\text{coef}}$ and $\ge_{\text{coef}}$ to denote the coefficient-wise inequalities. Let $H(z) := \prod_{j \notin \{i, i+1\}} T_u(p_j z)$, we can conclude:

$$S_{i+1} = (n - n_1)! \, [z^{n-n_1}] \, (T_u(p_i z) \cdot H(z)) \ge (n - n_1)! \, [z^{n-n_1}] \, (T_u(p_{i+1} z) \cdot H(z)) = S_i,$$

as well as

$$S_{i+1} = (n - n_1)! \, [z^{n-n_1}] \, (T_u(p_i z) \cdot H(z)) \le \left(\frac{p_i}{p_{i+1}}\right)^u (n - n_1)! \, [z^{n-n_1}] \, (T_u(p_{i+1} z) \cdot H(z)) = \left(\frac{p_i}{p_{i+1}}\right)^u S_i.$$

Thus, we obtain the desired results for $S_i$.

Next, for $\tilde{S}_i$: a key observation is that moving from $\tilde{S}_i$ to $\tilde{S}_{i+1}$ will not affect the corresponding weight $w(\mathbf{r})$ (as both $c_d'$ and $m(\mathbf{r})$ are fixed). We can thus apply the same idea in previous analysis and still get the results for $\tilde{S}_i$. Details are omitted.

$\square$

*Proof of Lemma A.2.* For any $L = 3, \cdots, K$: let $m := n - \sum_{t=1}^{L-1} n_t$, $u := n_{L-1}$, and $T_u(x) := \sum_{r=0}^{u} \frac{x^r}{r!}$. By the definition of $S_{i_1, \cdots, i_{L-1}}$, we have

$$S_{i_1, \cdots, i_{L-1}} = [z^m] \prod_{j \ne i_1, \cdots, i_{L-1}} T_u(p_j z).$$

For notation simplicity, we define $I := \{i_1, \cdots, i_{L-1}\}$ to be the index set. We take two indexes $a$ and $b$ such that $a < b$, $b \in I$ and $a \notin I$ (If there does not exists such $a$ and $b$, then we already have $I$ to be any permutation of $\{1, 2, \cdots, L-1\}$). Let $I' = (I \setminus \{b\}) \cup \{a\}$, we have

$$S_{I'} = [z^m] \left( T_u(p_a z) \cdot \prod_{j \notin I \setminus \{b\}} T_u(p_j z) \right)$$

and

$$S_I = [z^m] \left( T_u(p_b z) \cdot \prod_{j \notin I \setminus \{b\}} T_u(p_j z) \right).$$

As we have

$$T_u(p_b z) \le_{\text{coef}} T_u(p_a z) \le_{\text{coef}} \left(\frac{p_a}{p_b}\right)^u T_u(p_b z).$$

Taking the result back to $S_I$ and $S_{I'}$:

$$\left(\frac{p_b}{p_a}\right)^u \le \frac{S_{I'}}{S_I} \le 1.$$

Based on the result, assume that $b$ corresponds to the $i_t$-th index in $I$, we have:

$$\frac{A_{I'}}{A_I} = \left(\frac{p_a}{p_b}\right)^{n_t} \cdot \frac{S_{I'}}{S_I} \ge \left(\frac{p_a}{p_b}\right)^{n_t - u} \ge 1.$$

In another word, we can always replace some index $b$ in $I$ with an index $a < b$ that is not in $I$ and get a larger $A_{I'} \geq A_I$. Through this procedure, we can obtain the largest $A_I$ with $I$ to be any permutation of $\{1, 2, \cdots, L-1\}$.

As $S_I$ is invariant on the order of $i_1, i_2, \cdots, i_{L-1}$, then we only need to compare $\prod_{t=1}^{L-1} p_{i_t}^{n_t}$. By applying the rearrangement inequality, we can derive

$$\max_{i_2, \cdots, i_{L-1}} A_{1, i_2, \cdots, i_{L-1}} = A_{1, 2, \cdots, L-1}, \quad \max_{i_1 \neq 1} A_{i_1, i_2, \cdots, i_{L-1}} = A_{2, 1, 3, \cdots, L-1},$$

For $\tilde{S}_{i_1, \cdots, i_{L-1}}$, the result still holds as $w(\mathbf{r})$ is still invariant with the index set $I$. Thus we can conclude the proof.

$\square$

*Proof of Lemma A.3.* Recall the definition of $S_1^m = (1 - p_{1,m})^{t-t_1} \cdot \mathbb{P}_m^{(-1)}(\mathbf{r} \in \mathcal{R}_{t,t_1})$, where $\mathbb{P}_m^{(-1)}$ stands for the Multinomial probability distribution with parameter $t - t_1$ and $\mathbf{q}_m^{(-1)} := (q_{2,m}^{(-1)}, \cdots, q_{K,m}^{(-1)})$, for $q_{j,m}^{(-1)} := \dfrac{p_{j,m}}{1 - p_{1,m}}$ for any $j = 2, 3, \cdots, K$. We can translate the region $\mathcal{R}_{t,t_1}$ into:

$$\mathcal{F}_{\theta_t} := \left\{ \mathbf{r} \in \Delta^{K-1} : \max_{j \neq 1} r_j \leq \theta_t \right\}, \quad \theta_t := \frac{t_1}{t - t_1}.$$

Thus, it is sufficient to calculate $\mathbb{P}_m^{(-1)}(\mathbf{r} \in \mathcal{F}_{\theta_t})$ instead. Define $\rho := \dfrac{p_1}{1 - p_1}$. By applying the Sanov's theorem, we obtain:

$$\lim_{(t-t_1) \to \infty} \frac{1}{t - t_1} \cdot \log \mathbb{P}_m^{(-1)}(\mathbf{r} \in \mathcal{F}_{\theta_t}) = - \inf_{\mathbf{r} \in \mathcal{F}_\rho} D_{\mathrm{KL}}(\mathbf{r} || \mathbf{q}_m^{(-1)}).$$

As $\max_{j \neq 1} q_{j,m}^{(-1)} = q_{2,m}^{(-1)} = \dfrac{p_{2,m}}{1 - p_{1,m}}$. If $\rho \geq q_{2,m}^{(-1)}$, we can choose $\mathbf{r} = \mathbf{q}_m^{(-1)}$ and now $D_{\mathrm{KL}}(\mathbf{r} || \mathbf{q}_m^{(-1)}) = 0$; If $\rho < q_{2,m}^{(-1)}$, now the optimal $\mathbf{r} = (\rho, \frac{1-\rho}{1 - q_{2,m}^{(-1)}} \cdot q_{2,m}^{(-1)}, \frac{1-\rho}{1 - q_{2,m}^{(-1)}} \cdot q_{3,m}^{(-1)}, \cdots, \frac{1-\rho}{1 - q_{2,m}^{(-1)}} \cdot q_{K,m}^{(-1)})$. The corresponding KL divergence is

$$D_{\mathrm{KL}}(\mathbf{r} || \mathbf{q}_m^{(-1)}) = \rho \cdot \log \frac{\rho}{q_{2,m}^{(-1)}} + (1 - \rho) \cdot \log \frac{1 - \rho}{1 - q_{2,m}^{(-1)}} = D_{\mathrm{KL}}(\rho || q_{2,m}^{(-1)}).$$

Thus, we have its asymptotic rate of convergence to be:

$$\mathbb{P}_m^{(-1)}(\mathbf{r} \in \mathcal{F}_{\theta_t}) = \exp\left( -(t - t_1) \cdot D_{\mathrm{KL}}(\rho || q_{2,m}^{(-1)}) \cdot \mathbf{1}\{\rho < q_{2,m}^{(-1)}\} + o(t - t_1) \right),$$

and $B_1^m$ can be obtained thereafter. The same analysis holds for $B_2^m$.

For the terms of $\tilde{B}_1^m$ and $\tilde{B}_2^m$: by its definition, we can also rewrite $\tilde{S}_1^m$ as $\tilde{S}_1^m = (1 - p_{1,m})^{t-t_1} \cdot \mathbb{E}_m^{(-1)}[w(\mathbf{r}) \cdot \mathbf{1}\{\mathbf{r} \in \mathcal{F}_{\theta_t}\}]$ with expectation $\mathbb{E}_m^{(-1)}$ taken w.r.t. $\mathbf{r}$ following $\mathbb{P}_m^{(-1)}$. As a result,

$$\binom{c'_d + K}{c'_d}^{-1} \cdot \mathbb{P}_m^{(-1)}(\mathbf{r} \in \mathcal{F}_{\theta_t}) \leq \mathbb{E}_m^{(-1)}[w(\mathbf{r}) \cdot \mathbf{1}\{\mathbf{r} \in \mathcal{F}_{\theta_t}\}] \leq \mathbb{P}_m^{(-1)}(\mathbf{r} \in \mathcal{F}_{\theta_t}).$$

Besides, we notice that

$$\lim_{(t-t_1) \to \infty} \frac{1}{t - t_1} \cdot \log \binom{c'_d + K}{c'_d} = 0.$$

Therefore, the asymptotic rate of convergence in this case is the same as before, i.e., $\tilde{B}_1^m = B_1^m$. The same analysis also holds for $\tilde{B}_2^m$.

$\square$

# C. Appendix: Algorithm Details

---

**Algorithm 1** Optimal Stopping under $L$-Aggregated Posterior Approximation Scheme

---

**Input:** aggregation parameter $L \in \{2, \cdots, K\}$, confidence parameter $\delta \in (0, 1/2)$
Initialize $\mathcal{C}_0 = \emptyset$
**for** $n = 1, 2, \cdots$ **do**
    Sample LLM and get an observation
    Update $\mathcal{C}_n$ and map it to $\mathcal{C}_n^L$
    Compute $\mathbb{P}(H_1 \mid \mathcal{C}_n^L)$ following Algorithm 2
    **if** $\mathbb{P}(H_1 \mid \mathcal{C}_n^L) \geq 1 - \delta$ **then**
        **break**
    **end if**
**end for**
**Output:** the most frequent observation (with tie breaks arbitrarily)

---

Before illustrating the idea of efficiently calculating $\tilde{S}_\psi$ in Algorithm 2, we first introduce Algorithm 3 which is a simpler case for calculating $S_\psi$. Note that $S_\psi$ can be also expressed as:

$$S_\psi = \sum_{\mathbf{r}^{-\psi}} \frac{\bar{n}_L!}{\prod_{j \in [K] \setminus \psi} r_j!} \prod_{j \in [K] \setminus \psi} p_j^{r_j}$$
$$= \bar{n}_L! \, [z^{\bar{n}_L}] \left( \prod_{j \in [K] \setminus \psi_{L-1}} \sum_{r=0}^{\min\{n_{L-1}, \bar{n}_L\}} \frac{(p_j z)^r}{r!} \right), \tag{21}$$

where $[z^{\bar{n}_L}]$ denotes the coefficient of $z^{\bar{n}_L}$. Algorithm 3 efficiently computes this coefficient via dynamic programming by iteratively convolving the coefficients of the generating function in (21). Instead of enumerating all possible count partitions, which is computationally prohibitive, the algorithm treats the summation as a coefficient extraction problem from the product of exponential generating functions associated with each label. By iteratively performing discrete convolution on the coefficient arrays, Algorithm 3 computes the probability mass for the residual count $\bar{n}_L$ in polynomial time.

Then we turn to our Algorithm 2: Based on Algorithm 3, the main idea (difference) in Algorithm 2 is that we introduce a dummy variable $u$ to indicate that whether there exists label that is assigned with $v_d$. For each label $j \in J_\psi$ in the tail: we define the generating function

$$G_j(z, u) = \sum_{r=0}^{v_d - 1} \frac{(p_j z)^r}{r!} + u \cdot \frac{(p_j z)^{v_d}}{v_d!}.$$

Therefore, we have $[u^m z^{\bar{n}_L}] \prod_{j \in J_\psi} G_j(z, u)$ denoting the coefficients for the case where $\sum r_j = \bar{n}_L$ and the number of labels in the tail that hit the boundary $v_d$.

We further introduce the 2D table $A[m][t]$ that stores the coefficient that the total count of the tail is $t$ (corresponding to $[z^t]$) and the number of boundary hit is $m$ (corresponding to $[u^m]$), and $B[m][t]$ is the 2D table that serves as the cache for $A$ used for updating the next label. We first update $B$ until the total count of the tail is up to $v_d - 1$. Then, for the boundary hit case, where the total tail count is $v_d$, we further update $B[m+1][t+v_d]$ to increase the boundary hit count $m$, which corresponds to the term $u \cdot \frac{(p_j z)^{v_d}}{v_d!}$ in the generating function $G_j(z, u)$. The total computational complexity of these algorithms can be derived by counting the number of iterations directly.

---

**Algorithm 2** Computing $\tilde{S}_\psi$ for $L = 2, 3, \cdots, K$ using Dynamic Programming

---

**Input:** number of answers $K$; prior $(p_1, \cdots, p_K)$; aggregation parameter $L$; cutoff count $v_d$; head multiplicity $c'_d$; residual count $\bar{n}_L$; injective mapping $\psi \in \mathfrak{S}_{L(n)}$

Let $I_\psi \leftarrow \{\psi(1), \cdots, \psi(L(n))\}$; $J_\psi \leftarrow [K] \setminus I_\psi$; $N \leftarrow |J_\psi|$ (i.e., $N = K - L(n)$)

Let $m_{\min} \leftarrow \min\{N, \lfloor \bar{n}_L / v_d \rfloor\}$

Initialize coefficient arrays $A[0, \cdots, m_{\min}][0, \cdots, \bar{n}_L]$ and $B[0, \cdots, m_{\min}][0, \cdots, \bar{n}_L]$ by
    $A[0][0] \leftarrow 1$ and $A[m][t] \leftarrow 0$ for all other $(m, t)$; set $B[m][t] \leftarrow 0$ for all $(m, t)$

**for** each $j \in J_\psi$ **do**

    Compute $g_j[r] \leftarrow p_j^r / r!$ for all $r = 0, \cdots, v_d$         // coefficients of $G_j(z, u) = \sum_{r=0}^{v_d - 1} (p_j z)^r / r! + u(p_j z)^{v_d} / v_d!$

    Reset $B[m][t] \leftarrow 0$ for all $m = 0, \cdots, m_{\min}$ and $t = 0, \cdots, \bar{n}_L$

    **for** $m = 0$ **to** $m_{\min}$ **do**

        **for** $t = 0$ **to** $\bar{n}_L$ **do**

            **if** $A[m][t] = 0$ **then**

                **continue**

            **end if**

            $r_{\min} \leftarrow \min\{v_d - 1, \bar{n}_L - t\}$

            **for** $r = 0$ **to** $r_{\min}$ **do**

                $B[m][t + r] \leftarrow B[m][t + r] + A[m][t] \cdot g_j[r]$

            **end for**

            **if** $t + v_d \leq \bar{n}_L$ **and** $m + 1 \leq m_{\min}$ **then**

                $B[m + 1][t + v_d] \leftarrow B[m + 1][t + v_d] + A[m][t] \cdot g_j[v_d]$

            **end if**

         **end for**

        **end for**

    Set $A \leftarrow B$         // $A[m][t]$ now stores $[u^m z^t] \prod_{k \in J_\psi} G_k(z, u)$

**end for**

$\tilde{S}_\psi \leftarrow 0$

**for** $m = 0$ **to** $m_{\min}$ **do**

    $\tilde{S}_\psi \leftarrow \tilde{S}_\psi + \binom{c'_d + m}{c'_d}^{-1} \cdot A[m][\bar{n}_L]$

**end for**

$\tilde{S}_\psi \leftarrow \bar{n}_L! \cdot \tilde{S}_\psi$         // implements $\tilde{S}_\psi = \bar{n}_L! \sum_m \binom{c'_d + m}{c'_d}^{-1} [u^m z^{\bar{n}_L}] \prod_{j \in J_\psi} G_j(z, u)$

**Output:** $\widetilde{S}_\psi$

---

**Algorithm 3** Computing $S_\psi$ for $L = 2, 3, \cdots, K$ using Dynamic Programming

---

**Input:** number of answers $K$; prior $(p_1, \cdots, p_K)$; aggregation parameter $L$; cutoff count $v_d$; residual count $\bar{n}_L$; injective mapping $\psi \in \mathfrak{S}_{L(n)}$

Let $I_\psi \leftarrow \{\psi(1), \cdots, \psi(L(n))\}$; $J_{\psi_{L-1}} \leftarrow [K] \setminus I_\psi$; $R \leftarrow \min\{v_d, \bar{n}_L\}$

Initialize coefficient array $A[0, \cdots, \bar{n}_L]$ by $A[0] \leftarrow 1$ and $A[t] \leftarrow 0$ for all $t = 1, \cdots, \bar{n}_L$

**for** each $j \in J_{\psi_{L-1}}$ **do**

    Compute $g_j[r] \leftarrow p_j^r / r!$ for all $r = 0, \cdots, R$         // coefficients of $G_j(z) = \sum_{r=0}^{R} (p_j z)^r / r!$

    **for** $t = \bar{n}_L$ **down to** $1$ **do**

        $s \leftarrow 0$

        $r_{\min} \leftarrow \min\{R, t\}$

        **for** $r = 0$ **to** $r_{\min}$ **do**

            $s \leftarrow s + A[t - r] \cdot g_j[r]$

        **end for**

        $A[t] \leftarrow s$         // $A[t]$ now stores $[z^t] \prod_{k \in J_\psi} G_k(z)$

    **end for**

**end for**

$S_\psi \leftarrow \bar{n}_L! \cdot A[\bar{n}_L]$         // implements $S_\psi = \bar{n}_L! [z^{\bar{n}_L}] \prod_{j \in J_\psi} G_j(z)$

**Output:** $S_\psi$

---

# D. Appendix: Numerical Details

All experiments were conducted locally on a laptop with Apple M2 chip, 8-core ARM64 CPU, 8 GB memory in Python 3.7.

We provide some additional results for synthetic datasets with $\pi_1 = (0.6, 0.1, 0.1, 0.1, 0.05, 0.05)$ for $K = 6$; and $\pi_2 = (0.5, 0.2, 0.2, 0.05, 0.05)$ for $K = 5$. The following two tables provide results parallel to that in Table 1.

*Table 3.* Comparison across $L \in \{2, 3, 4\}$ over different thresholds $1 - \delta$ with prior $\pi_1$

| | $1 - \delta = 0.7$ | | $1 - \delta = 0.8$ | | $1 - \delta = 0.9$ | | $1 - \delta = 0.95$ | | $1 - \delta = 0.975$ | | $1 - \delta = 0.99$ | |
|---|---|---|---|---|---|---|---|---|---|---|---|---|
| Method | Mode Acc. | Num. Gen. | Mode Acc. | Num. Gen. | Mode Acc. | Num. Gen. | Mode Acc. | Num. Gen. | Mode Acc. | Num. Gen. | Mode Acc. | Num. Gen. |
| $L = 2$ | 86.8% | 3.44 | 87.6% | 3.59 | 93.3% | 4.68 | 97.8% | 6.37 | 98.8% | 7.42 | 99.5% | 8.87 |
| $L = 3$ | 83.9% | 3.07 | 87.1% | 3.48 | 93.7% | 4.63 | 96.8% | 5.68 | 99.0% | 7.36 | 99.4% | 7.98 |
| $L = 4$ | 83.9% | 3.07 | 87.0% | 3.46 | 93.7% | 4.63 | 96.9% | 5.71 | 99.0% | 7.36 | 99.4% | 7.98 |
| Exact | 83.9% | 3.07 | 87.0% | 3.46 | 93.7% | 4.63 | 96.8% | 5.68 | 99.0% | 7.36 | 99.4% | 7.99 |
| ASC | 60.6% | 1.00 | 94.2% | 5.16 | 99.5% | 9.73 | 99.9% | 13.36 | 100.0% | 16.90 | 100.0% | 22.18 |

*Table 4.* Comparison across $L \in \{2, 3, 4\}$ over different thresholds $1 - \delta$ with prior $\pi_2$

| | $1 - \delta = 0.7$ | | $1 - \delta = 0.8$ | | $1 - \delta = 0.9$ | | $1 - \delta = 0.95$ | | $1 - \delta = 0.975$ | | $1 - \delta = 0.99$ | |
|---|---|---|---|---|---|---|---|---|---|---|---|---|
| Method | Mode Acc. | Num. Gen. | Mode Acc. | Num. Gen. | Mode Acc. | Num. Gen. | Mode Acc. | Num. Gen. | Mode Acc. | Num. Gen. | Mode Acc. | Num. Gen. |
| $L = 2$ | 77.8% | 5.63 | 86.1% | 8.50 | 93.4% | 13.31 | 96.6% | 17.11 | 98.1% | 20.22 | 99.3% | 24.51 |
| $L = 3$ | 78.1% | 5.52 | 84.4% | 7.57 | 91.9% | 11.30 | 96.2% | 15.30 | 98.5% | 19.50 | 99.5% | 23.45 |
| $L = 4$ | 78.1% | 5.50 | 84.9% | 7.70 | 92.3% | 11.74 | 96.2% | 15.30 | 98.5% | 19.45 | 99.5% | 23.45 |
| Exact | 78.1% | 5.50 | 84.9% | 7.70 | 92.2% | 11.70 | 96.2% | 15.30 | 98.5% | 19.45 | 99.5% | 23.45 |
| ASC | 50.4% | 1.00 | 82.1% | 7.61 | 96.5% | 19.37 | 99.2% | 29.12 | 99.9% | 39.32 | 100.0% | 52.76 |

We also provide additional results for real-world data in FEVAL-TTC on different LLM models and on different datasets (DisambiguationQA, GSM8K). The results are presented as follows. Note that there may exist slight deficit in Mode Acc. that does not reach the threshold $1 - \delta$, which is due to the cutoff effect (as we only sample 100 answers for each question). Also notice that there might exist some cases where the efficiency of $L = 2$ even slightly better than $L = 3$ (in Table 5), which is again a consequence of the limited sampling size, as the finite number of generations (of 100 answers) can lead to a slight empirical bias.

*Table 5.* Comparison across methods and models under Dataset DisambiguationQA.

| | | $1 - \delta = 0.95$ | | | $1 - \delta = 0.975$ | | | $1 - \delta = 0.99$ | | |
|---|---|---|---|---|---|---|---|---|---|---|
| Model | Method | Ans. Acc. | Mode Acc. | Num. Gen. | Ans. Acc. | Mode Acc. | Num. Gen. | Ans. Acc. | Mode Acc. | Num. Gen. |
| | $L = 2$ (known prior) | 88.5% | 97.9% | 4.93 | 89.1% | 98.4% | 5.54 | 89.1% | 98.4% | 6.11 |
| | $L = 3$ (known prior) | 88.5% | 97.9% | 4.93 | 89.1% | 98.4% | 5.48 | 89.1% | 98.4% | 6.09 |
| Qwen-2.5-72B | $L = 2$ (uncertain prior) | 87.7% | 97.6% | 3.45 | 87.7% | 97.6% | 4.23 | 88.3% | 99.2% | 7.37 |
| | $L = 3$ (uncertain prior) | 87.7% | 97.6% | 3.41 | 87.7% | 97.6% | 4.19 | 88.3% | 99.2% | 7.44 |
| | ASC | 88.0% | 100.0% | 6.19 | 88.0% | 100.0% | 7.67 | 88.0% | 100.0% | 9.64 |
| | $L = 2$ (known prior) | 77.9% | 98.7% | 7.65 | 78.1% | 99.5% | 8.90 | 78.1% | 99.5% | 10.55 |
| | $L = 3$ (known prior) | 77.9% | 98.7% | 7.65 | 78.1% | 99.5% | 8.90 | 78.1% | 99.5% | 10.55 |
| GPT-4o mini | $L = 2$ (uncertain prior) | 77.3% | 97.6% | 7.24 | 77.1% | 98.4% | 9.46 | 77.9% | 99.2% | 14.14 |
| | $L = 3$ (uncertain prior) | 77.3% | 97.6% | 7.33 | 77.1% | 98.4% | 9.48 | 77.9% | 99.2% | 14.21 |
| | ASC | 78.7% | 100.0% | 8.05 | 78.7% | 100.0% | 10.24 | 78.7% | 100.0% | 12.56 |
| | $L = 2$ (known prior) | 75.7% | 97.6% | 3.87 | 76.3% | 99.2% | 4.78 | 76.3% | 99.2% | 5.68 |
| | $L = 3$ (known prior) | 76.0% | 97.3% | 3.60 | 76.3% | 99.2% | 4.74 | 76.3% | 99.2% | 5.63 |
| LLaMA-3.1-405B | $L = 2$ (uncertain prior) | 76.5% | 97.9% | 3.69 | 76.5% | 97.9% | 4.06 | 76.5% | 98.9% | 6.76 |
| | $L = 3$ (uncertain prior) | 76.8% | 97.6% | 3.35 | 76.8% | 97.6% | 3.50 | 76.5% | 98.9% | 6.59 |
| | ASC | 76.0% | 100.0% | 7.49 | 76.0% | 100.0% | 9.49 | 76.0% | 100.0% | 11.83 |

*Table 6.* Comparison across methods and models under Dataset GSM8K.

| Model | Method | $1 - \delta = 0.95$ | | | $1 - \delta = 0.975$ | | | $1 - \delta = 0.99$ | | |
|---|---|---|---|---|---|---|---|---|---|---|
| | | Ans. Acc. | Mode Acc. | Num. Gen. | Ans. Acc. | Mode Acc. | Num. Gen. | Ans. Acc. | Mode Acc. | Num. Gen. |
| Qwen-2.5-72B | $L = 2$ (known prior) | 97.1% | 99.8% | 2.03 | 97.2% | 99.8% | 2.19 | 97.2% | 99.8% | 2.51 |
| | $L = 3$ (known prior) | 97.1% | 99.7% | 2.00 | 97.2% | 99.8% | 2.15 | 97.2% | 99.8% | 2.44 |
| | $L = 2$ (uncertain prior) | 96.3% | 98.2% | 1.00 | 96.3% | 98.2% | 1.00 | 97.2% | 100.0% | 3.37 |
| | $L = 3$ (uncertain prior) | 96.3% | 98.2% | 1.00 | 96.3% | 98.2% | 1.00 | 97.2% | 100.0% | 2.95 |
| | ASC | 97.2% | 100.0% | 4.58 | 97.2% | 100.0% | 5.61 | 97.2% | 100.0% | 6.91 |
| GPT-4o mini | $L = 2$ (known prior) | 96.6% | 99.4% | 3.53 | 96.6% | 99.4% | 3.81 | 96.4% | 99.7% | 4.04 |
| | $L = 3$ (known prior) | 96.6% | 99.4% | 3.42 | 96.7% | 99.5% | 3.73 | 96.4% | 99.7% | 3.94 |
| | $L = 2$ (uncertain prior) | 94.3% | 96.5% | 1.00 | 96.6% | 99.1% | 3.98 | 96.6% | 99.0% | 4.41 |
| | $L = 3$ (uncertain prior) | 94.3% | 96.5% | 1.00 | 96.6% | 99.0% | 3.49 | 96.6% | 99.0% | 3.80 |
| | ASC | 96.8% | 99.6% | 5.33 | 96.8% | 99.6% | 6.64 | 96.4% | 100.0% | 8.27 |
| LLaMA-3.1-405B | $L = 2$ (known prior) | 97.0% | 99.8% | 1.95 | 97.0% | 99.8% | 2.09 | 97.2% | 100.0% | 2.31 |
| | $L = 3$ (known prior) | 97.0% | 99.8% | 1.88 | 97.1% | 99.9% | 2.04 | 97.2% | 100.0% | 2.25 |
| | $L = 2$ (uncertain prior) | 96.2% | 97.7% | 1.00 | 97.2% | 99.7% | 2.49 | 97.2% | 99.7% | 2.70 |
| | $L = 3$ (uncertain prior) | 96.2% | 97.7% | 1.00 | 97.2% | 99.7% | 2.42 | 97.2% | 99.7% | 2.64 |
| | ASC | 97.2% | 100.0% | 4.50 | 97.2% | 100.0% | 5.78 | 97.2% | 100.0% | 7.02 |

