# OpenReview forum: "Optimal Bayesian Stopping for Efficient Inference of Consistent LLM Answers"
_ICML.cc/2026/Conference — ICML 2026 regular_

### Official Review · Reviewer_p7vs · 2026-03-11

**Soundness:** 4
**Presentation:** 2
**Significance:** 3
**Originality:** 3
**Overall Recommendation:** 4
**Confidence:** 3

**Summary:**

This paper presents a practical finding for LLM error remediation via self-consistency: it is sufficient to establish a stopping criterion based on the occurrence counts of only the two most frequent answer modes plus an aggregate of all other results. The authors substantiate this "L=3" rule by proving its asymptotic optimality through a theoretical "known prior" framework before extending it to the more realistic "uncertain prior" setting.

**Compliance With Llm Reviewing Policy:**

Affirmed.

**Key Questions For Authors:**

none

**Limitations:**

yes

**Strengths And Weaknesses:**

# Strengths
## Non-obvious "Sweet Spot" Discovery
The paper presents a counter-intuitive result that tracking only the top two most frequent answers ($L=3$) is sufficient to achieve the same asymptotic efficiency as tracking every possible unique answer.

## Thorough Empirical Validation
 The authors conducted extensive testing across multiple LLM families (LLaMA, Qwen, DeepSeek, GPT) using the standardized FEVAL-TTC benchmark to ensure results weren't model-specific.

# Weaknesses
## Focus on an Impractical "Known-Prior"
Significant portions of the theoretical foundation rely on a "known-prior" case—the assumption that the exact probability distribution of answers exists and is measureable—which is almost never true in real-time inference on new queries.

## Brittle Observation Model (String Identity):
The framework relies on the premise that answers can be categorized by checking if they are "identical or not". In real-world applications, this creates a heavy and potentially unreliable dependency on:
External Extraction Layers: The system requires a pre-processing step (like regex or LLM-based parsing) to standardize strings before the stopping algorithm can group them.
Lack of Semantic Awareness: The observation model cannot natively recognize that "42" and "forty-two" are the same result unless they are normalized first, making it prone to over-sampling due to minor formatting variations.

---

> ### Author Rebuttal · Authors · 2026-03-28
>
> We appreciate the reviewer's comments. Below we provide a point-to-point response for the comments on the weakness.
>
> **Response to Weakness 1 on Impractical ''Known-Prior'':** We sincerely thank the reviewer for raising this important point. We address this concern by clarifying how the ''known prior'' can be practically estimated, highlighting its theoretical necessity, and explaining how our ''uncertain prior'' framework may help for real-world queries.
>
> 1. Practicality and Theoretical Value of the Known Prior:
> We admit that assuming a known prior for a new query without any historical data is impractical, while it can be reasonably estimated in real-world systems with sufficient historical data provided. For a new question, we can perform an embedding-based similarity search against the historical dataset. By identifying the most similar historical question, we can directly adopt its empirical answer distribution (obtained via offline repeated sampling in the dataset) as the known prior for the new query. Furthermore, analyzing the known-prior case is theoretically indispensable. It yields clean and insightful theoretical results, establishing the fundamental limits and optimal stopping behavior that ground our entire framework.
>
> 2. Resolving Impracticality via the Uncertain Prior: To explicitly address the reality that a perfect prior is rarely available, we introduced the uncertain prior framework. This approach drops the assumption of an exact prior and instead assumes the true prior belongs to a known candidate set $\Pi^M$. In practice (as implemented in Section 5.3), we partition our dataset into training and testing sets and extract the empirical answer distributions of every question in the training set to construct the candidate set $\Pi^M$, and then run our algorithm on the testing queries. The uncertain prior framework not only provides its own interesting theoretical guarantees but also demonstrates good numerical performance. In our experiments, we utilized a very coarse, uninformative hyper-prior by simply setting $\lambda_m \equiv 1/M$ (assuming every candidate prior in $\Pi^M$ is equally likely). Even under this rough, unoptimized setting, our algorithm can still significantly outperform the prior-free ASC method, yielding massive sample reductions while maintaining comparable accuracy.
>
> **Response to Weakness 2 on String Identity:** We thank the reviewer for this sharp and insightful observation. We completely agree that without semantic awareness, pure string matching can be brittle. We note that: (1) As the reviewer correctly anticipated, our code implementation indeed includes a data preprocessing and normalization step. For instance, using regular expressions, text variations like ''36/2'' are standardized into the numeric format ''18'' before the stopping algorithm evaluates them. This is a standard and necessary engineering practice in LLM evaluation pipelines. (2) Besides, for the reasoning tasks we evaluate (e.g., mathematical reasoning and multiple-choice questions), determining whether two final answers are identical is straightforward. In these scenarios, extracting and normalizing the final answer (e.g., the numerical value or the option letter) is highly deterministic and exceptionally accurate. This reliable preprocessing ensures that semantically identical answers are stably mapped to the same element, effectively minimizing the risk of over-sampling caused by minor formatting variations.

---

### Official Review · Reviewer_52NU · 2026-03-11

**Soundness:** 3
**Presentation:** 2
**Significance:** 3
**Originality:** 2
**Overall Recommendation:** 4
**Confidence:** 3

**Summary:**

Summary:

The paper tackles the problem of determining when to stop sampling LLM responses during self-consistency decoding. Current approaches use a fixed number of samples, which is inefficient. While Adaptive Self-Consistency (ASC) methods exist, they ignore prior information about answer distributions that could be learned from the LLM's performance on similar problems.

Main Contributions:

Bayesian Framework: The authors formulate adaptive stopping as a sequential hypothesis testing problem that incorporates informative priors about the LLM's answer distribution on similar queries. They derive an optimal Bayesian sampling rule, though exact posterior computation has factorial complexity O(K!).

L-Aggregated Posterior Approximation: To address computational intractability, they introduce an efficient approximation scheme that tracks only the L-1 most frequent answers while aggregating the rest. This reduces complexity from O(K!) to O(KL).

Theoretical Results: The paper proves that L=3 (tracking the top two answers plus "others") achieves asymptotic optimality—matching the sample efficiency of the exact posterior while being computationally tractable. They show this strictly outperforms prior-free baselines.

Extension to Uncertain Priors: The framework is extended to scenarios where the exact prior is unknown but belongs to a candidate set learned from related problems, with theoretical guarantees for this hierarchical setting.

Empirical Validation: Experiments on synthetic data and the FEVAL-TTC benchmark demonstrate that L=3 achieves similar accuracy to baselines while using significantly fewer samples (e.g., 80%+ reduction in some cases), validating the theoretical findings.

The key insight is that tracking just the top two answer frequencies captures sufficient information for optimal stopping decisions, providing an efficient "sweet spot" between statistical and computational complexity.

**Compliance With Llm Reviewing Policy:**

Affirmed.

**Final Justification:**

The authors have addressed all the concerns that I had raised in the review. The significance of the paper will depend on availability of the code and reproducibility of the results. It is also very important that all the limitations are clearly discussed in the paper, instead of false promising with L=3 and not delivering. The authors seem to understand these concerns but I do not know if ICML has a way to make sure that these are addressed/delivered. I would move the score up ONLY under the assumption that the authors would keep to their promise.

**Key Questions For Authors:**

When should one use known vs. uncertain prior framework?
No discussion of how to collect historical data to estimate priors.

Are you able to provide exact framework with code and steps to reproduce the results you have mentioned in section 5.3. Real-World Evaluation on FEVAL-TTC?

**Limitations:**

Constructive suggestions:

Add a dedicated "Limitations" section before the conclusion.
 Explicitly state assumptions (e.g., stationary question distributions, availability of historical data) Discuss failure modes and when practitioners should/shouldn't use this method .
Acknowledge the experimental scope is limited to certain task types

Potential negative impacts that should be discussed:

Premature stopping incentives: Organizations motivated by cost reduction might use overly aggressive confidence thresholds, sacrificing accuracy for speed. This could be particularly harmful in high-stakes applications (medical, legal, financial decisions).

Amplification of biases: If the prior is estimated from biased historical data, the method could amplify those biases by stopping earlier when answers align with historical patterns.

Unequal access: Sophisticated prior estimation infrastructure may only be available to well-resourced organizations, potentially widening the gap between large and small AI deployers.

False confidence: Providing calibrated stopping rules might give users false confidence in LLM outputs, when the fundamental issue is that LLMs can confidently produce wrong answers.

**Strengths And Weaknesses:**

Strengths
Soundness
The paper demonstrates strong theoretical rigor:

Rigorous theoretical framework: The formulation as optimal Bayesian stopping for mode identification is mathematically sound, with careful treatment of the observation model (count-of-counts representation) and posterior computation.

Complete proofs: Theorems 3.2 and 4.1 provide asymptotic characterizations with detailed proofs in the appendix. The analysis carefully handles edge cases (e.g., when ρ < ρ_2^(-1)) and uses appropriate concentration inequalities (Hoeffding, Chernoff bounds).

Non-trivial theoretical insight: The result that L=3 achieves asymptotic optimality (Theorem 3.2) is surprising and well-substantiated. The proof technique using "sandwich bounds" and decomposition of log-likelihood ratios is sound.

Empirical validation: Experiments on both synthetic and real-world data (FEVAL-TTC) align well with theoretical predictions, showing the L=3 approximation matches exact posterior performance at high confidence levels.

Presentation
The paper is generally well-written with clear structure:

Clear motivation: The introduction effectively motivates the problem—reducing sampling costs in self-consistency while maintaining accuracy guarantees.

Good use of examples: Example 2.1 helps illustrate the count-of-counts representation, making abstract concepts concrete.

Related work mentioned: Section 1.1 positions the work well relative to ASC methods and mode identification literature.

Appropriate use of appendices: Complex proofs and algorithm details are relegated to appendices.

However, readability of the paper is questionable, with text consisting of heavy theoretical notations, formatting errors and inconsistencies.

Significance
The work addresses a practically important problem:

Real computational savings: Experiments show 80%+ reductions in samples needed in some cases, which translates directly to cost and latency reductions in LLM inference.

Principled approach: Provides a theoretically grounded alternative to heuristic stopping rules, with calibrated confidence guarantees.

Scalability: The O(K^L) complexity makes the method practical for real-world deployment where K can be large.

Broader applicability: The framework applies to any scenario where mode identification from expensive samples is needed, beyond just LLMs.

Originality
The paper makes several novel contributions:

New problem formulation: Introduces "Bayesian mode identification with informative priors" where the prior is over answer frequencies (not answer identities), avoiding the triviality of standard Bayesian mode identification.

Novel approximation scheme: The L-aggregated posterior is a creative solution balancing statistical and computational efficiency.

Unexpected theoretical result: The sufficiency of L=3 for asymptotic optimality is non-obvious and provides actionable guidance.

Extension to uncertain priors: The hierarchical framework (Section 4) naturally handles practical scenarios where exact priors are unknown.

Weaknesses
Soundness
Limited experimental scope:

Only tested on multiple-choice style questions (CommonsenseQA, AQUA) where K is naturally bounded. Performance on open-ended tasks with large K is unclear.
The FEVAL-TTC benchmark uses cached responses, so real-time performance characteristics aren't validated.
Only 5 repetitions for some experiments (Table 2) seems limited for statistical significance.
Gap between theory and practice:

Asymptotic results require δ → 0, but experiments use moderate values (0.95-0.99). Non-asymptotic finite-sample guarantees would strengthen practical applicability.
The assumption that π can be reliably estimated from historical data isn't validated—what if question distributions shift?
Theorem 4.1 shows L=2 can degrade badly with uncertain priors, but experimental section doesn't explore when this failure mode occurs in practice.
Missing ablations:

No analysis of sensitivity to hyperprior λ_m (uniform assumption may be suboptimal).
How does performance vary with the size M of the candidate prior set?
What happens when the true prior isn't in Π_M?
Computational complexity claims:

While O(K^L) is stated, actual wall-clock time comparisons are limited (only Table 1). For large K scenarios, is L=3 actually fast enough?
Algorithm 2's dynamic programming may have poor cache performance—practical considerations aren't discussed.
Presentation
Dense notation:

Formatting errors and inconsistensies in 2.2. Exact Posterior Calculation.
The paper introduces heavy notation (S_i, S̃i, S^m_i, ρ, ρ^(-1){2,m}, etc.) that becomes difficult to track, especially in Section 4.
The count-of-counts representation C_n = {(v_i, c_i)} takes effort to parse; more intuitive exposition would help.
Insufficient intuition:

Why does L=3 suffice? The paper states it captures "occurrence % and difference with second-most-frequent" but doesn't provide deep intuition for why this is the critical threshold.
The connection between w(r) correction term and tie-breaking (Remark 3.1) is confusing and could use better explanation.
The mathematical notations are not lucid neither is the there a flow in the supporting text, which makes understanding of the paper challenging.
Experimental presentation:

Tables 1-2 report many metrics, making key takeaways hard to extract. Plots showing stopping time vs. confidence level would be clearer.
The "uncertain prior" construction (70/30 train/test split) isn't well-motivated—why this particular setup?
Missing practical guidance:

How should practitioners choose L in non-asymptotic regimes?
When should one use known vs. uncertain prior framework?
No discussion of how to collect historical data to estimate priors.
Significance
Limited scope of improvements:

On highly consistent models (Table 2: Qwen72B25 on CommonsenseQA), ASC already uses ~6-9 samples. The marginal benefit of 3-6 samples is modest.
Answer accuracy differences between methods are minimal, suggesting the mode may not always be correct anyway.
Comparison to simpler baselines:

Only compares to Beta-stopping ASC. What about simpler heuristics like "stop after k consecutive same answers" or entropy-based stopping?
No comparison to recent work like BEACON (Wan et al., 2025b) cited in related work.
Deployment considerations:

Requires maintaining prior distributions for different question types—infrastructure burden not discussed.
Cold-start problem: what happens for novel question types without historical data?
Disconnect with answer accuracy:

Mode accuracy ≠ answer accuracy (Table 2: 99.4% mode accuracy but only 83% answer accuracy). If the mode is often wrong, why optimize for finding it?
The paper doesn't address when self-consistency itself fails.
Originality
Incremental over ASC:

The core idea—using priors to improve adaptive stopping—is natural. The main novelty is the L-aggregation scheme and theoretical analysis.
Prior work (Wang et al., 2025) already explored "difficulty-aware" approaches, though not formalized as Bayesian stopping.
Limited algorithmic novelty:

The L-aggregation is essentially a compression technique. While the L=3 sufficiency result is nice, the method itself is straightforward.
Dynamic programming for computing generating function coefficients (Algorithm 2-3) is standard technique.
Narrow theoretical contribution:

The asymptotic optimality result is specific to this formulation. It's unclear if insights transfer to other stopping problems.
The "mode identification with frequency priors" framing, while clever, is quite specialized.
Additional Concerns
Reproducibility: While the paper uses FEVAL-TTC benchmark, code availability isn't mentioned. The complex DP algorithms would benefit from open implementation.

Statistical testing: No confidence intervals or significance tests reported for experimental comparisons.

Though 5.3. Real-World Evaluation on FEVAL-TTC is there, the authors fail to state how to verify these results or the numbers in the appendix (supplementary code expected).

---

> ### Author Rebuttal · Authors · 2026-03-28
>
> We appreciate the Reviewer for the detailed feedback. Due to the strict space limitations, we have carefully synthesized your feedback into the following four core themes. Should any secondary questions remain unaddressed by this high-level grouping, we warmly welcome further discussion during the subsequent discussion period.
>
> **1. Empirical Breadth and Baselines:**(1) Performance on Open-Ended Tasks and Statistical Significance:
> We first clarify that even for the datasets currently tested (i.e., CommonsenseQA and AQUA), the potential number of distinct answers K is not naturally bounded, and the LLM can still generate a wide variety of distinct responses. To demonstrate our algorithm's effectiveness on open-ended tasks and to ensure robust statistical significance, we conducted additional experiments on the GSM8K and SVAMP datasets. In these new tests, we increased the number of repeated runs to 20 and provided the corresponding confidence intervals. Please refer to the tables in our anonymous link https://anonymous.4open.science/r/Paper-0100/. (2) Comparison to Other Benchmarks: We chose ASC as our baseline because the classic ASC paper has already established its empirical superiority over simpler heuristics like Best-of-N. Regarding more recent methods, such as BEACON, they typically rely on additional assumptions and resources. Therefore, comparison with such methods is not applicable to our framework.
>
> **2. Practical Deployment and Prior Estimation:**(1) Cold-Start Problem: Regarding the specific practical deployment and prior estimation of our algorithm, please refer to our response to Reviewer k8XZ (Question 1). We fully acknowledge that deploying our method requires historical data and inherently faces a cold-start problem. Without historical data, our method naturally degrades to a prior-free approach identical to standard ASC. The core benefit of our framework lies precisely in its ability to leverage historical data (i.e., prior) to drastically reduce sampling costs. Following your suggestion, we will add a dedicated ''Limitations'' discussion in the revised manuscript to explicitly point out the cold-start problem and data collection dependencies. (2) Known vs. Uncertain Prior (When to use which): Theoretically, when the prior is perfectly known, the Known-Prior algorithm should be used because it achieves a superior asymptotic rate of convergence. From a practical deployment perspective, if we have high confidence in the estimated prior, the Known-Prior algorithm is the optimal choice. However, as you pointed out, relying on a specific estimated prior introduces challenges regarding estimation robustness. While a deep analysis of estimation robustness falls outside the primary theoretical focus of this paper, we agree it is a critical practical consideration. We will highlight this trade-off and list prior estimation robustness as an open question in our new ''Limitations'' discussion.
>
> **3. The Gap between Mode Accuracy and Ground Truth:**(1) Why Optimize for the Mode: Optimizing for the mode is the fundamental objective of the standard ASC framework. By designing our algorithm to optimally identify the mode, we are aligning exactly with the proven objective of ASC, but achieving it with a significantly reduced sampling cost. (2) False Confidence Issue: Regarding the concern of false confidence when the actual correct answer is a non-mode answer, please refer to our detailed response to Reviewer k8XZ (Question 3).
>
> **4. Intuition and Reproducibility:** (1) Intuition for why $L=3$ suffices: Detecting the mode requires finding the most frequent answer. While $L=2$ only tracks the most frequent answer versus all others, $L=3$ tracks both the most frequent and the second most frequent answers. The gap between these two is the key: a larger gap gives us higher confidence that the top answer is the true mode. Because $L=3$ successfully captures this crucial gap, it can therefore achieve asymptotic optimality. (2) Reproducibility and Code Availability: We apologize for not including the code in our initial supplementary material. We assure you that all code is fully available and will be made open-source upon publication. Due to the ICML rebuttal policy, which only allows anonymous links for tables and figures, we cannot provide an anonymous link to our code repository at this stage. Regarding the DP algorithm (i.e., Algorithm 2) in the appendix, its purpose is simply to efficiently compute $\tilde{S}_{\psi}$ for the posterior. The algorithm itself is not complex; it merely trades memory space for computational speed. Regarding the concern about the poor cache performance of Algorithm 2, we would like to clarify that the worst-case space complexity of the algorithm is bounded by $O(Kn)$, where $n$ is the total number of repeated samplings. In practical applications, $n$ typically does not exceed 100. Therefore, the concern regarding poor cache performance does not manifest in practice.

---

> > ### Author Rebuttal · Reviewer_52NU · 2026-04-02
> >
> > The reviewers are promising to provide open source code. If they can add a clear Limitations section with the points given in the rebuttal, it will enhance the originality and significance of the paper for practitioners. Also please take care of formatting as some equations (eg. eq 2 ) overflow text width.

---

> > > ### Author Response · Authors · 2026-04-02
> > >
> > > We thank the reviewer for their time and constructive feedback throughout the review process, and for recognizing our rebuttal. We are fully committed to incorporating the final suggestions by releasing our open-source code and checking and fixing the formatting issues in the final manuscript.
> > >
> > > Regarding the suggestion to include a "Limitations" section, we have drafted the following paragraph based on the rebuttal discussions, which will be added to the final manuscript as well.
> > >
> > > **Limitations and Future Work**
> > > While our framework can help achieve significant sampling cost savings for LLM test-time scaling, we acknowledge several practical limitations and open questions: (i) Cold-start issue: Our framework leverages historical data to estimate prior information, thereby improving sampling efficiency. For completely new domains or scenarios without any collected data, it reverts to the standard prior-free ASC baseline. (ii) Prior sensitivity: In cases where the estimated prior is biased, the theoretical performance guarantees on the robustness of our algorithms remain an open question to be explored. (iii) Non-mode ground truth: In scenarios where the LLM's inherent mode answer is not actually the ground truth, how to appropriately extend our framework to address this discrepancy presents an interesting question for future research.

---

### Official Review · Reviewer_k8XZ · 2026-03-14

**Soundness:** 4
**Presentation:** 4
**Significance:** 3
**Originality:** 4
**Overall Recommendation:** 5
**Confidence:** 4

**Summary:**

This paper is about making self-consistency cheaper at test time. The setup is: instead of always sampling a fixed number of CoT responses and majority-voting at the end, they cast stopping as a Bayesian sequential decision problem and stop once the current leading answer has enough posterior probability. The main technical issue is that the exact posterior is combinatorial and basically intractable when there are many possible answers, so they introduce an L-aggregated posterior that only tracks the top few answer counts rather than the full histogram. The key claim is that tracking just the top two answers plus the rest aggregated (L = 3) is already enough: it gives the same asymptotic stopping efficiency as the exact Bayesian rule, while being much cheaper to compute. Empirically, on synthetic setups and FEVAL-TTC replay experiments, the method seems to reach similar mode / answer accuracy with fewer samples than prior-free adaptive self-consistency, especially when the prior is known or reasonably estimated from related questions.

**Compliance With Llm Reviewing Policy:**

Affirmed.

**Key Questions For Authors:**

1. Could the author help me understand how the method may have different assumptions on the prior? Is the proposed method assuming the true prior in the candidate set? how does it impact the proposed method and how does it impact the comparison with standard SC methods?

2. Can you evaluate the stopping rule with live decoding rather than replayed/subsampled trajectories?

3. Could the author provide more insights on how to handle the cases where the non-mode answers can be correct?

4. Just curious, could the method be modified to increase the diversity of the answer? asking because I'm curious if similar methods can be used to improve the RL rollout diversity.

**Limitations:**

yes

**Strengths And Weaknesses:**

Strengths

* The main technical contribution is pretty nice. The exact posterior is combinatorial and impractical, and the L-aggregated posterior is a simple approximation that still keeps the right signal for stopping.

* I like the “three is all you need” result. Showing that tracking the top answer, the second answer, and everything else is enough for asymptotic optimality is a crisp and memorable takeaway.

* The paper has a good theory/algorithm tradeoff story. It is not just proposing an approximation; it explains why coarser aggregation reduces computation, and why L ≥ 3 still preserves the asymptotic stopping efficiency of the exact rule.

* The setup is also practically relevant. Using prior information from similar questions to reduce self-consistency cost makes sense, especially since fixed-budget SC is often wasteful in practice.

* Empirically, the paper seems to validate the intended benefit: fewer samples while keeping answer accuracy competitive.

Weakness

* The “real-world” evaluation is still replay-based rather than live. FEVAL-TTC gives cached generations, and the paper then simulates new trajectories by subsampling from the empirical distribution, so it never really tests end-to-end stopping with fresh decoding.

* Lacking end-to-end efficiency profiling numbers to tell the wall-clock saving. I guess the author could have discussed how the mordern hardware design impact the acceleration comparison--I assume mv voting is more friendly when people have multipel gpus available?

*

---

> ### Author Rebuttal · Authors · 2026-03-28
>
> We appreciate the reviewer's comments and the insightful questions provided. Below we provide a point-to-point response for your key questions.
>
> **Response to Question 1 on Assumptions on the Prior and Comparison with ASC:**  We clarify our prior assumptions, their practical implementations, and their theoretical and empirical relation to standard ASC below:
>
> 1. Assumptions and Practical Implementations:
> (1) Known Prior (Section 3): Assumes the exact prior is known. In practice, for a new question, we can perform an embedding-based similarity search against a historical dataset and directly adopt the empirical answer distribution of the most similar historical question as the ''known'' prior. (2) Uncertain Prior (Section 4): Assumes the true prior belongs to a candidate set $\Pi^M$. In our experiments (Section 5.3), we construct $\Pi^M$ using the empirical answer distributions of all training set questions, and evaluate our algorithm on the testing set.
>
> 2. Comparison with Prior-Free ASC:
> (1) Theoretical Connection: When our candidate prior set becomes sufficiently large and uninformative (where we have $M \to \infty$), our asymptotic stopping rate converges exactly to that of ASC. Thus, ASC can be viewed as the limiting case of our framework when no effective prior knowledge is available. (2) Empirical Superiority: For the uncertain prior, we used a very coarse uniform hyper-prior ($\lambda_m \equiv 1/M$). Even without optimizing this setting (e.g., by weighting candidates based on similarity), our algorithm still yields massive sample reductions over ASC while maintaining comparable accuracy.
>
> **Response to Question 2 on Live Decoding:** We thank the reviewer for the practical question.  We utilized the cached FEVAL-TTC dataset primarily for two practical reasons: (1) ensuring strict reproducibility free from API fluctuations, and (2) avoiding the prohibitively high compute cost of massive live LLM sampling. As our paper is fundamentally a theoretical work establishing an optimal Bayesian stopping framework, we believe validating our method on standardized cached data is sufficient at this stage. Deploying this algorithm in a live decoding system is a valuable direction for future real-world implementation.
>
> **Response to Question 3 on Non-mode Answer:** Thank you for your question. We have tested on the CommonsenseQA dataset using the Qwen72B model and it shows that the exact mode matches the true answer 87.55\% of the time, and the probability that the true answer falls within the top-2 most frequent answers increases to 93.28\%. This explains why our method achieves an accuracy level comparable to ASC. When our algorithm makes a mistake, it most commonly misidentifies the second mode as the true mode. Because this second mode is frequently the correct answer, our method is inherently robust to cases where the true mode is incorrect, allowing us to maintain a comparable accuracy level to ASC. Additional insights to handle correct non-mode answers include: (1) returning the top-2 answers instead of just the mode, and (2) aggregating results from multiple LLMs to mitigate single-model errors.
>
> **Response to Question 4 on Rollout Diversity:** This is a very interesting idea. Our current framework is designed to identify the mode with the minimum number of samples, which inherently reduces sampling diversity. To ensure sampling diversity, which we interpret as guaranteeing a given ''coverage level'' (the distinct answers we have sampled so far account for at least a certain percentage of the total true probability mass), a potential implementation is to use the Bayesian framework in reverse. That is, given a target coverage level and a confidence threshold, the algorithm would continue sampling to discover new answers and mathematically stop only when the desired coverage level is reached with given confidence threshold. Our framework may help in reducing the total sampling cost for the problem in such a way.

---

> > ### Author Rebuttal · Reviewer_k8XZ · 2026-04-02
> >
> > My questions are resolved. Nice work. I look forward to seeing its broader applications & improvements to make it more fault-tolerant. I will keep my positive score.

---

> > > ### Author Response · Authors · 2026-04-03
> > >
> > > We sincerely thank you for your time and encouraging feedback, and we look forward to exploring those broader applications and improvements in our future work.

---

### Decision · Program_Chairs · 2026-04-30

**Decision:**

Accept (regular)

**Comment:**

This paper studies how to reduce the sampling cost of self-consistency decoding in LLMs by formulating the stopping decision as a Bayesian sequential problem. To address the intractability of the exact posterior, the authors propose an L-aggregated posterior that tracks only the most frequent answers, and show that using (L=3) (top-2 answers plus the rest) achieves asymptotically optimal stopping efficiency. The framework is further extended to uncertain priors, and experiments on synthetic data and FEVAL-TTC demonstrate that the method can significantly reduce the number of samples while maintaining comparable accuracy.

Reviewers generally find the paper technically strong and well-motivated. They highlight the solid theoretical foundation, the elegant and practical L-aggregation approximation, and the insightful “L=3 suffices” result. Empirical results are also seen as supportive of the claims. At the same time, reviewers note several limitations, including reliance on replay-based evaluation rather than live decoding, limited experimental breadth (e.g., open-ended settings), practical concerns around prior estimation, and some issues with clarity and presentation. Additional concerns include limited discussion of real-world efficiency and deployment considerations.

In the rebuttal, the authors provided clarifications on prior assumptions and their practical estimation, explained the use of cached data and positioned live decoding as future work, and added further experimental evidence and implementation details. They also addressed concerns about non-mode correctness, reproducibility, and limitations. Reviewers acknowledged that their main concerns were adequately resolved.

Overall, the paper presents a solid and meaningful contribution with strong theoretical insights and promising empirical results. I recommend acceptance.